# The sensitivity of simulated streamflow to individual hydrologic processes across North America

Juliane Mai 🔟 [1✉], James R. Craig 🔟 [1], Bryan A. Tolson 🔟 [1] & Richard Arsenault 🔟 [2]

Streamflow sensitivity to different hydrologic processes varies in both space and time. This sensitivity is traditionally evaluated for the parameters specific to a given hydrologic model simulating streamflow. In this study, we apply a novel analysis over more than 3000 basins across North America considering a blended hydrologic model structure, which includes not only parametric, but also structural uncertainties. This enables seamless quantification of model process sensitivities and parameter sensitivities across a continuous set of models. It also leads to high-level conclusions about the importance of water cycle components on streamflow predictions, such as quickflow being the most sensitive process for streamflow simulations across the North American continent. The results of the 3000 basins are used to derive an approximation of sensitivities based on physiographic and climatologic data without the need to perform expensive sensitivity analyses. Detailed spatio-temporal inputs and results are shared through an interactive website.

[1] Department of Civil and Environmental Engineering, University of Waterloo, Waterloo, ON, Canada. [2] Department of Construction Engineering, École de technologie supérieure, Montreal, QC, Canada. ✉email: juliane.mai@uwaterloo.ca

Hydrologic models are widely used in applications that are important for society such as flood prediction[1–6], drought monitoring[7–10], infrastructure design[11–13], and reservoir management[14–16]. This wide variety of such applications, coupled with the diversity of climatic and physiographic regions and the underlying complexity of hydrologic processes is leading to increasing complexity among these models[17–19]. Further developments and improvements of hydrologic models are essential to advance the understanding of hydrologic processes and ensure greater model realism[20–23]. One way such improvements can be ensured is by carrying out model evaluations taking advantage of information theory and newly available datasets[24–29]. Sensitivity analysis (SA) is a well-established tool to guide such model assessments[30], navigate model development[31], and identify the most critical relationships within a system[32–34]. SA is based on large sets of model runs identifying the most sensitive parameters of a model and is thus a general method that can be applied to any kind of model that contains unknown parameter estimates[35–37]. Note that parameters can be (traditional) model parameters, multiplicative factors to perturb input forcings, or parameters to weight between different options, among others.

Notwithstanding the repute of SA as a tool, there are several challenges limiting the transferability and insights of individual analyses. Four such challenges, here highlighted for hydrologic applications, are:

**Model parameters only:** SAs traditionally only estimate the sensitivities of model parameters on streamflow[35,38] or sensitivity indices of parameters on components or processes of the water cycle[34], rather than quantifying the sensitivity of streamflow to hydrologic processes, which limits insights in process understanding. Parameter-based analyses are model-specific and rarely lead to conclusions that can be transferred to other models.

**Dependence on location:** SAs are based on thousands of model runs, which makes them computationally very expensive. They are therefore usually only carried out for individual locations[39–42], and this limits the transferability of the obtained results to other locations.

**Dependence on model structure:** SAs are generally performed for individual models, which further limits the generality of conclusions[30,33,43].

**Data sharing and re-usability:** In-depth SA results, especially when applied to multiple locations using complex models, are usually not shared in an easily accessible way due to the amount and complexity of the data, which makes it challenging to obtain or compile information of interest for further usage[44–46].

This work applies the extended Sobol' Sensitivity Analysis (xSSA) method of Mai et al.[47] to a set of more than 3000 modelled locations across North America. The novelty of the xSSA method is that it generates process sensitivities in addition to the traditionally derived parameter sensitivities (addressing limitation 1). The continental-scale deployment allows for conclusions that hold over large domains (addressing limitation 2). The method is applied to the "Blended Model" for streamflow simulation introduced by Mai et al.[47], which enables a seamless analysis of model parameters and model structures, reducing the dependence of the results on specific modeling choices (addressing limitation 3). The input data, model setups, and SA results are shared on a map-based interactive website for members of the hydrologic research community to browse, explore, download and use for their specific regions of interest (addressing limitation 4).

## Results
Figure 1 provides an overview of the analyses performed in this study. The HYSETS database[48] is screened for watersheds with adequate overall data availability and catchment size, and the

blended hydrological model is developed, calibrated, and tested in validation. Models with adequate performance in the calibration period are subjected to xSSA analyses, enabling the deduction of functional relationships between basin attributes and the sensitivity of hydrologic processes at any location.

**Preliminary calibration and validation of the blended model.** Figure 2 shows the results of the preliminary basin-wise calibration of the 3826 basins with enough observed data in the calibration period (January 1991 to December 2010), as well as the performance of the 3005 calibrated basins which have enough data available during the validation period (January 1971 to December 1990). The median daily streamflow Nash-Sutcliffe efficiency (NSE) is 0.73 in calibration and 0.64 in validation. This is comparable to the performance of other models applied across the continental US (CONUS). For instance, Rakovec et al.[49] reported a median NSE of 0.72/0.66 (calibration/validation) for the mHM model[50,51] over 492 CONUS basins. Mizukami et al.[52] and Rakovec et al.[49] reported median NSEs of 0.61/0.57 (calibration/validation) for the VIC model[53,54] applied to the same basins, and[55] reported median NSEs of 0.7 to 0.75/0.6 to 0.65 (calibration/validation) using the SAC-SMA/Snow-17 model[56,57] over 671 CONUS basins, with variations due to meteorological forcings. The weaker performance of the blended model during validation over the high plains and desert southwest (seen in Fig. 2) is consistent with all the above-mentioned models. Regions with NSE performances lower than 0.5 are considered to be unreliable, and the basins are not included in the analyses to follow. The 3316 basins with an NSE of at least 0.5 are used going forward from here on. The selection of basins based on this threshold is applied for calibration performance rather than validation performance as the calibration period is the period used for the sensitivity analysis.

It is important to note that the results of this calibration exercise are used to (1) exclude clearly low quality models from further analysis and (2) to demonstrate the basic adequacy of the models for simulating streamflow over the range of simulated conditions. While a more elaborate calibration study may improve individual optimal model performances, it is unlikely to yield improved global sensitivity estimates. The exclusion of low quality models is admittedly not standard practice in sensitivity analyses. However, ensuring that models are able to represent physical processes is critical to confidently conclude on the spatial behavior of process sensitivities. Detailed results for all basins calibrated and validated, including the calibrated model setups, can be found on the website[58] associated with this publication.

**Spatial variation in hydrologic process sensitivity.** Figure 3 shows the variance-weighted total Sobol' sensitivity index $ST_i^w$, a metric representing the sensitivity of streamflow to variations of hydrologic processes. The variance weighting of timesteps for temporal aggregation is chosen to increase the importance of timesteps with high flows, which, in general, is a favored scheme in hydrologic applications[33,59]. The total Sobol' index includes interactions between parameters and between processes. The results show clear spatial patterns of the importance of the hydrologic processes with regard to streamflow. The patterns are in agreement with hydrologic reasoning; for example, snow balance sensitivity is high in mountainous and northern regions and potential melt is only sensitive where snow occurs. The sensitivity analysis results were determined to be robust to changes in specified parameter ranges, with negligible (< 0.0135) changes to process sensitivities based an analysis using a subset of 150 randomly selected basins (results not shown).

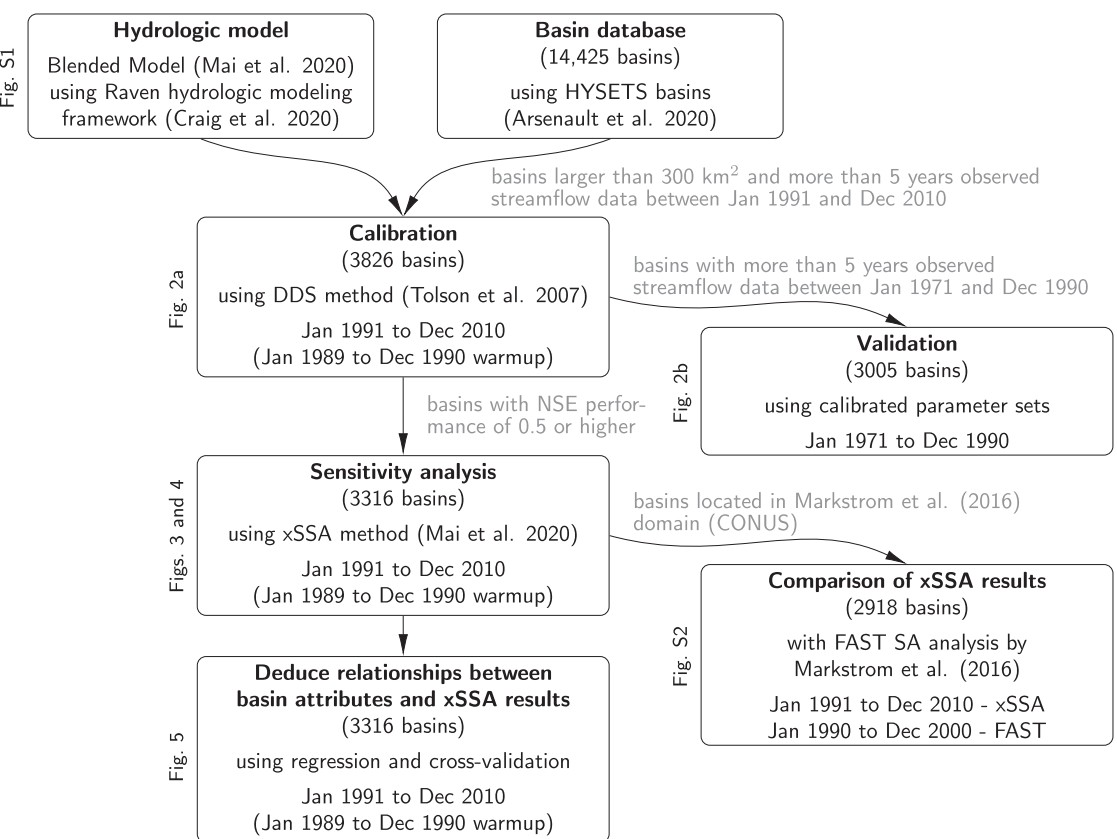

**Fig. 1 Flowchart of experiments and analyses.** All experiments performed in this work are listed, including the number of basins on which the analyses are based, the methods used, and the time periods over which the analyses are performed. The reasons for reducing the number of basins from the original 14,425 in the HYSETS database to 3316 for the xSSA sensitivity analysis and the deduction of relationships between basin characteristics and xSSA sensitivity results are added as gray labels to the arrows. The figures displaying the main results of each analysis are added as labels to the left of each box describing the analysis. The methods and results of the comparison of the xSSA results with another study can be found in the Supplementary Material (Fig. S2).

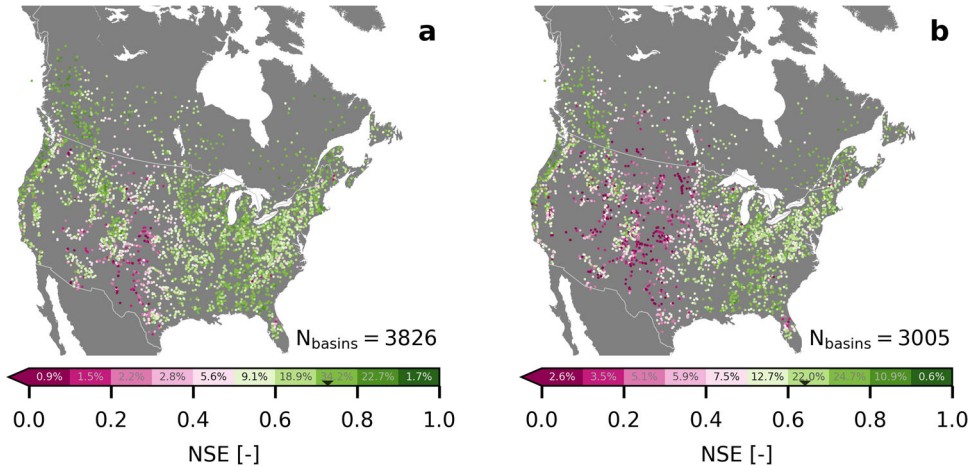

**Fig. 2 Preliminary calibration and validation results of blended model.** The performance with respect to the daily Nash-Sutcliffe efficiency (NSE) of the blended model during (**a**) calibration period (January 1991 to December 2010) and (**b**) validation period (January 1971 to December 1990). In total, 3826 basins with more than five years of streamflow observation data available during the calibration periods are calibrated. The 3005 basins that also had more than five years of data available during the validation period are validated. Each basin is represented on the map by its location of the streamflow gauge station (colored dots), while the color indicates the NSE performance. The distribution of basin performances is indicated on the colorbar. The black triangles (▼) in the colorbars mark the median NSE performance of 0.73 for calibration and 0.64 for validation. The 3316 basins with an NSE performance of 0.5 or higher during calibration are used for the remaining analyses of this study.

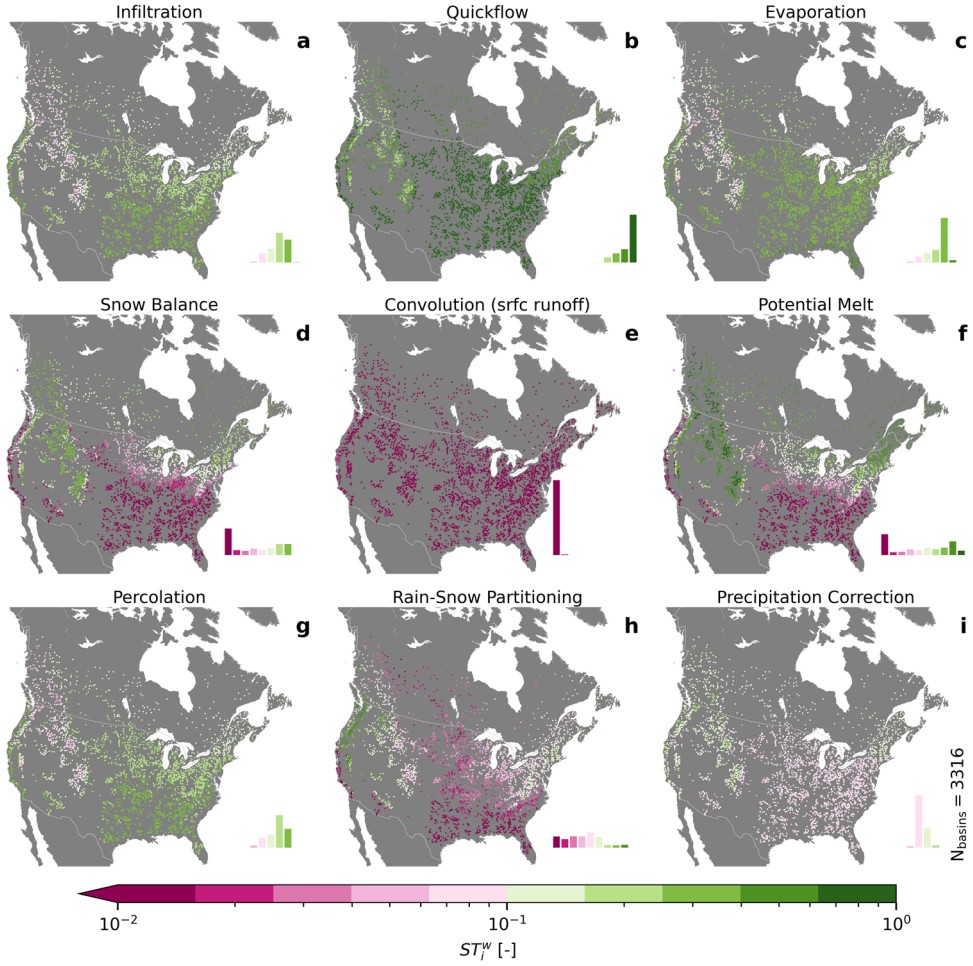

**Fig. 3 Importance of hydrologic processes for streamflow simulations.** The variance-weighted total Sobol' sensitivity $ST_i^w$ regarding simulated streamflow for the nine hydrologic model components, i.e.: (**a**) infiltration, (**b**) quickflow, (**c**) evaporation, (**d**) snow balance, (**e**) convolution of surface runoff, (**f**) potential melt, (**g**) percolation, (**h**) rain-snow partitioning, and (**i**) precipitation correction. The results are shown for the 3316 basins that have a calibrated Nash-Sutcliffe efficiency equal to or better than 0.5 in the calibration period (January 1991 to December 2010). The two additional processes of Baseflow and Convolution (delayed runoff) are analyzed but not displayed as all basins show a sensitivity of less than 0.01 (lower limit of colorbar). Please note that the colorbar is logarithmic in order to allow for a better distinction of small sensitivity estimates. Green colors indicate a large importance of the respective process on streamflow simulations while pink colors indicate a weak impact. The location of the dots in each panel marks the location of the outlet of each basin, which coincides with a streamflow gauging station. The histograms of sensitivity values are added as an inset to each map. The histogram bins are the same as used for the colorbar.

Overall, quickflow (Fig. 3b) is the most important process for streamflow simulations, with a median $ST_i^w$ of 0.736 across the 3316 basins analyzed. It exhibits large sensitivities, especially in the Eastern United States, which is covered mostly in temperate broadleaf and mixed forests. Infiltration, evaporation, and percolation (Fig. 3a, c, and g, respectively) are processes of secondary importance across North America, but especially in the Eastern US, with median $ST_i^w$ of 0.300, 0.21, and 0.202, respectively. In the Rocky Mountains and in higher latitudes (starting over the Great Lakes), the model's potential melt, precipitation correction and snow balance components (Fig. 3f, i, and d) are the most important, with an overall median $ST_i^w$ of 0.102, 0.087, and 0.047, based on the 3316 basins analyzed. On the West Coast, rain-snow partitioning (Fig. 3h) becomes more influential, with an overall median $ST_i^w$ of 0.054. The convolution of surface runoff (Fig. 3e), a process which controls the timing (rather than magnitude) of flow, is found to be insensitive throughout, with a median $ST_i^w$ of 0.003. The convolution of delayed runoff and baseflow are even less sensitive, with an $ST_i^w$ below 0.01 in each basin. Both latter processes are thus not shown

in Fig. 3. It is not surprising that these three processes (convolution of surface and delayed runoff and baseflow) are less sensitive since the sensitivity analysis assesses the variability in streamflow magnitudes rather than its timing.

The lack of sensitivity studies conducted over large domains and the novelty of the sensitivity method presented by Mai et al.[47] in estimating sensitivities of processes rather than model parameters lead to challenges in comparing results to those of previous studies. However, a large-scale sensitivity study across the continental US was performed by Markstrom et al.[34] using the US Geological Survey's Precipitation-Runoff Modeling System (PRMS)[60]. Their study derives the sensitivity of model parameters on eight hydrologic processes outputs. We compared the first-order mean (rather than time-dependent) sensitivities of runoff equivalent to $S_i$ in the Markstrom et al.[60] study, and the results are presented in the Supplementary Material (Fig. S2). While the sensitivity metrics are (strictly speaking) not equivalent, a correlation test yields a Pearson correlation coefficient of 0.88, and the sensitivities exhibit similar spatial trends as those presented here. The analysis herein furthermore

represents an improvement upon the Markstrom et al.[34] study, as it provides insights averaged over several model structures, is based on time-dependent sensitivities, and includes sensitivity estimates accounting for parameter interactions ($ST_i$) besides the first-order effects ($S_i$).

The sensitivity of hydrologic processes at the continental scale has been highlighted here due to its novelty. However, the xSSA analysis does not only derive the sensitivity of the hydrologic processes but also derives the sensitivity of the parameters and process options of the blended model regarding the time series of simulated streamflow. The detailed results, including maps of average main and total Sobol' sensitivity indices ($S_i^m$ and $ST_i^m$) and variance-weighted main and total Sobol' sensitivity indices ($S_i^w$ and $ST_i^w$) for all parameters, process options, and processes, as well as the according summary plots, are available on the website associated with this publication[61].

**Regional variation in transient hydrologic process sensitivity.** Figure 4 shows the time-dependent sensitivities of each process clustered by similarity. The time-dependent total sensitivities $ST_i$ regarding the simulated streamflow $Q(t)$ are averaged for each day of the year for the 20-year simulation period from January 1991 to December 2010. Each plot shown is a representative example basin of each clustered region (Fig. 4a, h). The regions are obtained by a c-means fuzzy clustering[62] based on the three climate indicators, namely, aridity, seasonality, and fraction of precipitation as snow. The example basin is then selected as being the closest to the cluster centroid based on the climate indicators (not the spatial centroid). Although the clustering is unsupervised, the identified regions align well with known physiographic and ecological regions, and named here according to this alignment.

In all regions, quickflow (medium blue) is the most important process, as already deduced from the time-averaged sensitivities in Fig. 3. Evaporation (dark blue) is also important, but mostly during low-flow periods (consistent with low weights represented by the black line). Potential melt (i.e., incoming energy, in orange) and snow balance (i.e., snow ablation processes, in green) are mostly important during cold months, and especially during the freshet (if it exists). Baseflow (dark green) is only visible in some regions (Fig. 4c, e, f), and only during severe low-flow periods. The two convolution processes controlling the timing of runoff (light green and yellow) are important almost nowhere, except in regions with large freshet events in April and May (Fig. 4g, h). The overall results are consistent with the time-aggregated sensitivities. However, the variation in sensitivity of processes throughout the year yields more detailed insights into the seasonal variability in hydrological process influence across basins.

To elaborate, the prototypical responses in the Coastal and Interior Plains (Fig. 4a) and Arid Regions and Florida regions (Fig. 4b) are almost similar. The two regions show near-constant sensitivities throughout the year for almost all active processes (i.e., infiltration, quickflow, evaporation, potential melt, percolation, and rain-snow partitioning). Potential melt (orange) and rain-snow partitioning (medium red) exhibit an increased importance during the winter months (December to February and December to March), while no overall strong streamflow variability is present throughout the year (as seen from the timestep weight in black).

The prototypical basin for the Mediterranean California and Temperate Sierra region (Fig. 4c) shows a higher streamflow during winter months and an elevated importance of rain-snow partitioning (medium red) during that time period, while baseflow (dark green) becomes relevant during the low-flow periods.

The fourth cluster of mainly Temperate Broadleaf and Mixed Forests (Fig. 4d) region with mild freshet and regular mid-winter melt events is the first basin showing an impact of snow balance (medium green) on the streamflow simulations during the winter months (January to March).

The basin representing the Boreal Forest region (Fig. 4e) shows that baseflow (dark green) becomes important during the severe low-flow winter period (December to mid-February) and potential melt (orange), as well as snow balance (green), are highly elevated during the freshet (March to April).

The example basin for the Temperate Coniferous Forests region (Fig. 4f) has large amounts of precipitation during the winter and spring months (large weights between December and April), but almost none is snow (blue channel of RGB of climate index is small). The summer months (June to October) of this catchment are fairly dry (low weight equals low flow). This leads to two sensitivity regimes: during winter, snow balance (green), potential melt (orange) and rain-snow partitioning (medium red) are elevated, while during summer, evaporation (dark blue), percolation (light red) and baseflow (dark green) become more important. The importance of infiltration (light blue) and quickflow (medium blue), however, is almost constant throughout the entire year. The latter shows a slightly decreased sensitivity during the melt period (March and April) when other processes become more important.

The freshet, with its extreme high-flow periods (April to May), is even more pronounced for the two examples in the Strongly Seasonal and Snow-Dominated Regions (Fig. 4g) and Montane Cordillera (Fig. 4h) snow-dominated clusters. The latter has a longer flattened high-flow period, while the first generally peaks in April and early May. In both basins, snow balance (green) and potential melt (orange) have an increased sensitivity during the high-flow freshet period, while during the low-flow summer months, evaporation (dark blue) is gaining importance. In all cases, the results of the sensitivity analysis performed here are generally consistent with our hydrological expectations of these landform types.

The temporal sensitivity patterns are catchment-specific, and vary between basins within a region. The temporal sensitivities of the 3316 analyzed basins can be viewed on the interactive map on the website associated with this publication[63].

**Estimating process sensitivity directly from basin attributes.** The sensitivities generated through xSSA were regressed against basin characteristics such as basin area and climatology (e.g., annual total precipitation based on the entire period of available forcings from 1950 to 2010), in order to assess whether sensitivities were readily determinable without the extensive analysis performed here. Figure 5 shows the predictability of the process sensitivities $ST_i^w$ using these regressions, with each functional relationship for each process using one predictor unless the use of two predictors increased the adjusted coefficient of determination $R_{adj}^2$ by at least 0.05. The regression model selection was guided by performance in predictive mode assessed in cross-validation experiments. The regressed relationships are given in Table 1. The relationships are based on regressions using the entire set of 3316 basins (the skill of that regression is given as $R_{adj}^2$ in the table). To obtain a measure of how sensitive these functions are to the choice of training basins, the basin set is split into 100 random subsets of basins, with two-thirds used for calibration and one-third for validation. The functions shown in Table 1 are then fitted and subsequently validated on the basins not used for training. For each of the 100 trials, the $R_{adj}^2$ and the mean absolute error $MAE$ between the true sensitivities derived using the xSSA analysis (see Fig. 3) and the predicted sensitivity based on the

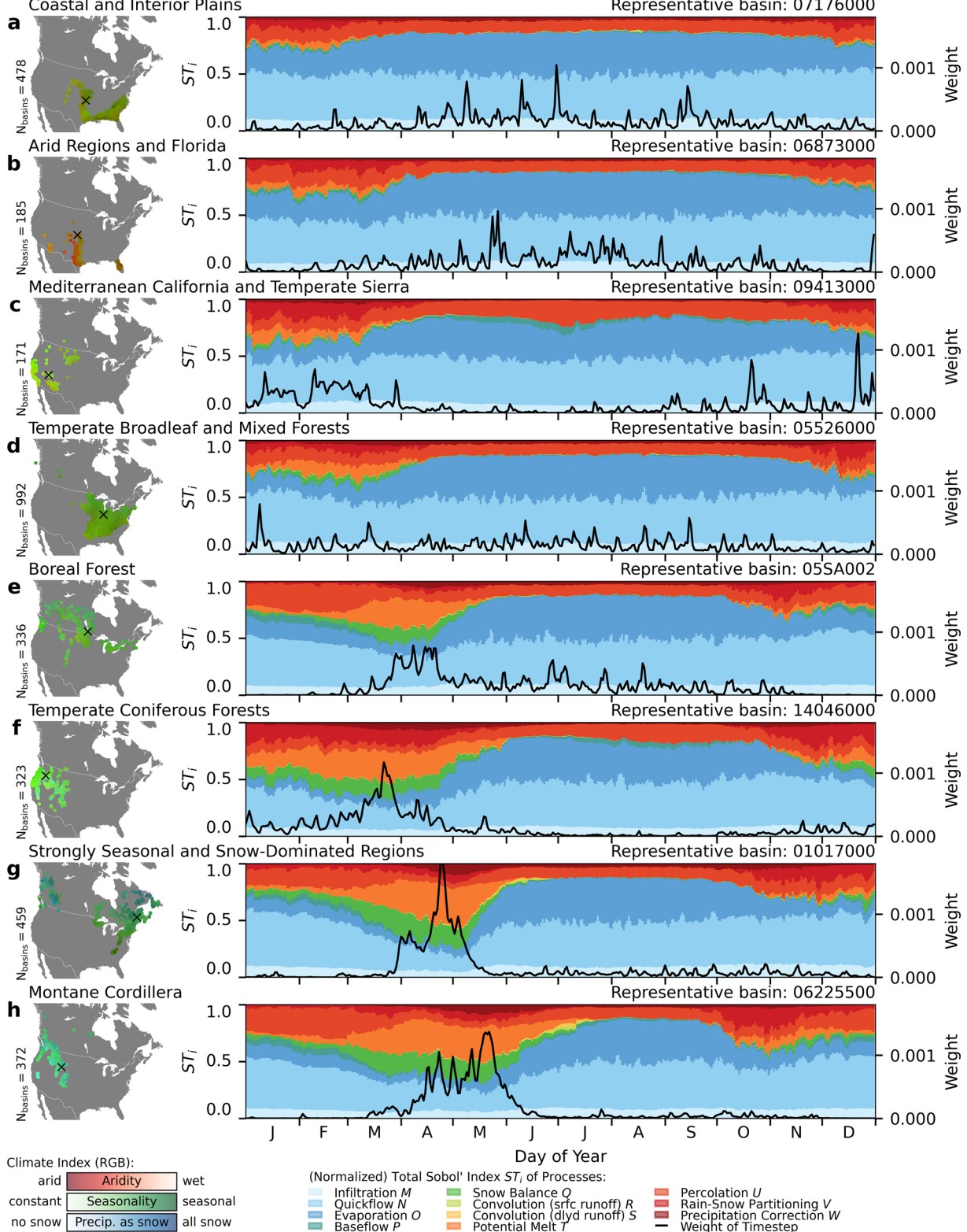

**Fig. 4 Sensitivity of processes over time at selected locations.** The importance of the eleven hydrologic processes (colorbars in right column panels) on each day of the year (average over 20 simulated years) are shown for eight example locations (**a**–**h**). These example locations are indicated by the USGS or WSC gauge station identifier added as a label at the top of the right column panels. These eight representative locations are the centers of eight clusters based on the three climate indicators. The clustering and example basins are derived to show the spatial variation of the process sensitivities over time in various climatic regions. Each cluster is named using the major physiographic or ecohydrologic region it covers (label on top of left column panels). For reference, the average weight of each day of the year based on the average simulated streamflow is displayed (black line in right column plots). These weights are used to derive time-aggregated sensitivities (see Fig. 3) from the time-dependent sensitivity indices displayed here in order to increase the weight of timesteps with high flows as compared to low-flow timesteps.

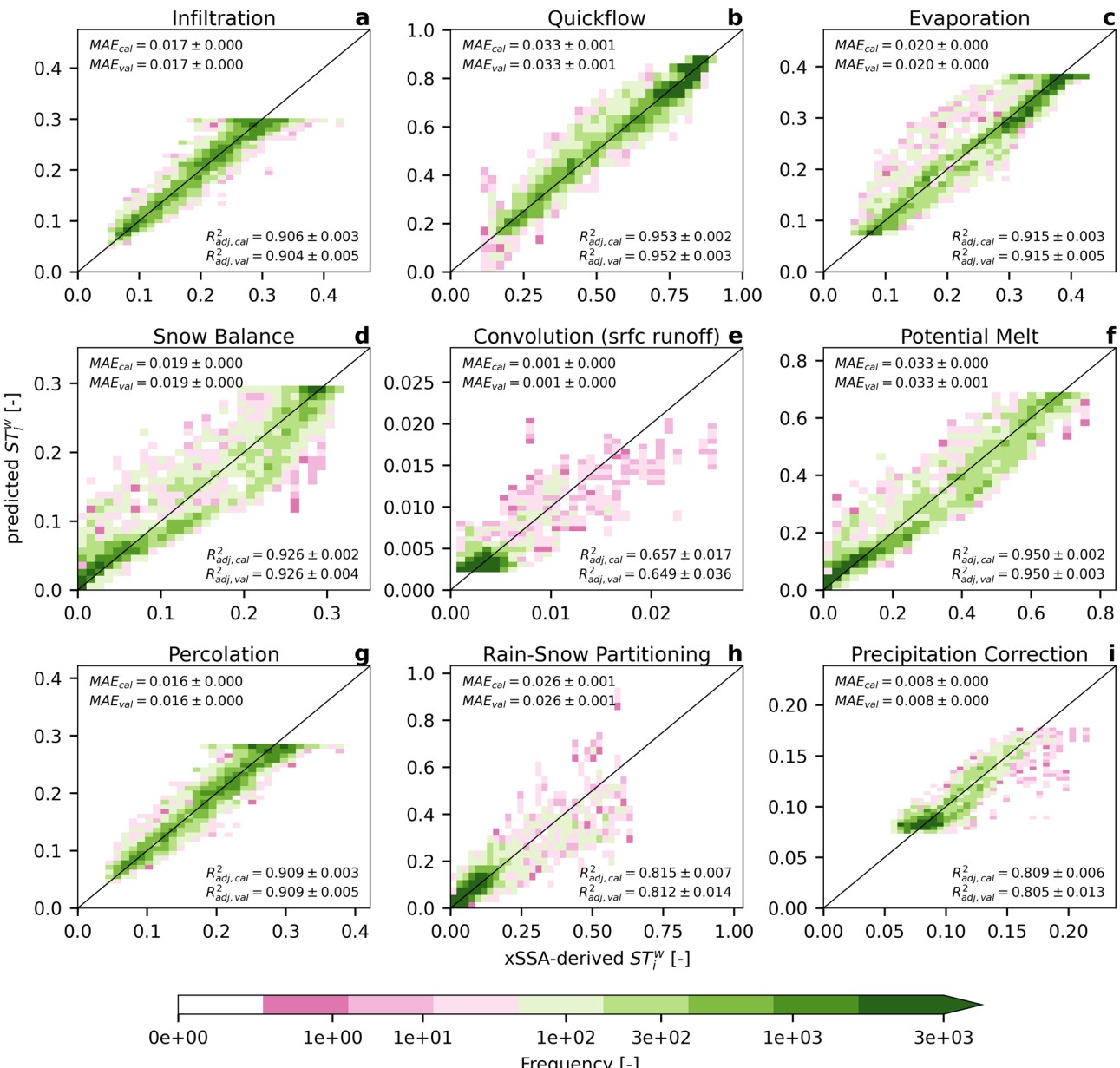

**Fig. 5 Predicted sensitivity index estimates using functional relationships based on basin characteristics.** The total Sobol' sensitivity indices of hydrologic processes on streamflow are first estimated using the xSSA method (Fig. 3; x-axis), after which they are predicted through basin-specific characteristics only (y-axis) and using the relations in Table 1. Each panel shows the results for one hydrologic process, i.e.: (**a**) infiltration, (**b**) quickflow, (**c**) evaporation, (**d**) snow balance, (**e**) convolution of surface runoff, (**f**) potential melt, (**g**) percolation, (**h**) rain-snow partitioning, and (**i**) precipitation correction. The mean and standard deviation of the Adjusted Coefficient of Determination $R^2_{adj}$ and the mean absolute error (*MAE*) are reported. Both indexes are derived between (i) xSSA-derived indices and (ii) the predicted indices and added as labels for calibration (*cal*) and validation (*val*) sets. The color indicates the density of samples. The samples shown here are the samples of the 100 validation experiments.

regression are derived. The individual panels of the figure show the 2D histogram of 100 sets of validation basins (i.e., $100 \times 1/3 \times 3316 > 100,000$ data points). The two processes Baseflow and Convolution of delayed runoff are not analyzed as their sensitivity is lower than 0.01 for all basins.

All panels show that the predicted sensitivities are in close agreement with those derived by the xSSA analysis, indicating that just a handful of basic variables may be sufficient for describing the process sensitivity at most sites. The results are consistent between calibration *cal* and validation *val* for both fit metrics $R^2_{adj}$ and *MAE*. Further, the 100 regression trials yield very consistent results, as indicated by the small standard deviations of the four metrics. The adjusted coefficients of determination $R^2_{adj}$

are above 0.80 for all processes in calibration and validation, except for Convolution of Surface Runoff (Fig. 5e; $R^2_{adj} \approx 0.65$), which is the least sensitive process. Only 88 of the 3316 basins show sensitivities larger than 0.01 for this process, which could explain the reduced overall predictability. In contrast, the very sensitive Infiltration (Fig. 5a), Quickflow (Fig. 5b), Evaporation (Fig. 5c), and Percolation (Fig. 5g) processes, in particular, show reliably large frequencies (dark green color) along the expected 1:1 line for large sensitivities. This, in turn, means that the processes with large sensitivity and importance can be identified and quantified reliably even without performing the expensive sensitivity analysis itself. The Potential melt (Fig. 5f), Precipitation correction (Fig. 5i), and Snow balance (Fig. 5d) processes,

**Table 1 Deduced functional relationships using basin attributes to estimate process sensitivities.**

| Process | Pred. x | Pred. y | Estimated functional relationship $\widetilde{ST}_i^w = f(x, y)$ | Adj. Coeff. of Determ. $R_{adj}^2(\overline{ST}_i^w, ST_i^w)$ |
|---|---|---|---|---|
| Infiltration | $f_S$ | – | $0.2985 - 0.8113x + 1.0443x^2 - 0.5203x^3$ | 0.905 |
| Quickflow | $f_S$ | $\Sigma P$ | $0.9044 - 3.4122x^2 + 2.6596x^3 - 4.771 \times 10^{-5}y - 0.001153xy + 0.001870x^2y$ | 0.953 |
| Evaporation | $f_S$ | – | $0.3835 - 0.7713x + 0.1945x^2 + 0.3964x^3$ | 0.915 |
| Snow Balance | $f_S$ | – | $-0.009051 + 0.4161x + 1.3796x^2 - 2.0661x^3$ | 0.926 |
| Convolution (srfc runoff) | $\overline{T}$ | $f_{cold}$ | $0.06531 - 0.00639x + 1.645 \times 10^{-4}x^2 - 0.0008165y + 4.104 \times 10^{-5}xy + 2.6508 \times 10^{-6}y^2$ | 0.657 |
| Potential Melt | $f_S$ | – | $-0.01494 + 0.8236x + 2.4265x^2 - 3.1523x^3$ | 0.950 |
| Percolation | $f_S$ | – | $0.2829 - 0.7275x + 0.8276x^2 - 0.3582x^3$ | 0.909 |
| Rain-Snow Partitioning | $f_S$ | $f_{cold}$ | $-0.001715 + 3.7710x - 0.002150y - 0.03638xy + 1.302 \times 10^{-5}y^2 + 8.753 \times 10^{-5}xy^2$ | 0.815 |
| Precipitation Correction | $f_S$ | $l_{m,r}$ | $0.07343 + 0.5829x^2 - 1.0106x^3 + 0.01122y^3 - 0.1677xy^2 + 0.5264x^2y$ | 0.809 |

Note. Results of the regression analysis identifying the functional relationships between predictors and the variance-weighted total Sobol' sensitivities $ST_i^w$ based on the results of the 3316 basins analyzed by xSSA. The regression is performed using polynomials with one predictor (up to degree three with at most six coefficients) and polynomials with two predictors (up to degree three with at most five coefficients). The one-predictor polynomial is used unless the use of a pair of predictors led to an improvement of the adjusted coefficients of determination $R_{adj}^2$ by at least 0.05. The basin characteristics used as predictors are derived from geophysical attributes and the meteorology available for each basin (1950 to 2010). The adjusted coefficients of determination $R_{adj}^2$ between the xSSA-derived sensitivities $ST_i^w$ and the predicted sensitivities $\overline{ST}_i$ are given in the last column. Description of predictors: $f_S$ is the fraction of precipitation that is snow [mm/mm], $\Sigma P$ is the annual sum of precipitation [mm], $\overline{T}$ is the annual average temperature in [°C], $f_{cold}$ is the average annual number of days below 0 °C, $l_{m,r}$ is the measure of seasonality ranging between [0,2].

already identified to be important in the Rocky Mountains and the higher latitudes starting over the Great Lakes (see Fig. 3), are also reliably quantified, even for the rarer large sensitivity indices.

It is revealing to examine the most reliable sensitivity predictors from Table 1, which summarizes the best predictors identified by the regression analysis. The two processes not listed—Baseflow and Convolution of Delayed Runoff—have a total Sobol' sensitivity of 0.01 or less in each of the basins. The predictor with the largest influence is the fraction of precipitation as snow $f_S$, which is used as a predictor in eight of the nine relationships. This is in close agreement with the findings by Konapala et al.[64], who also identified the fraction of snow as a key characteristic to explain runoff signals in their study based on an information theory approach. The four processes that benefit from adding a second predictor are Quickflow, Convolution of Surface Runoff, Rain-Snow Partitioning and Precipitation Correction. While we do not want to read too much into the results for the almost insensitive Convolution of Surface Runoff process, the addition of the annual sum of precipitation $\Sigma P$ to Quickflow, the average annual number of days below 0 °C $f_{cold}$ to Rain-snow Partitioning, and seasonality $I_{m,r}$ to Precipitation Correction as a second predictor improved the $R_{adj}^2$ significantly (from 0.884 to 0.953, from 0.618 to 0.815, and from 0.650 to 0.809, respectively). These three secondary predictors are consistent with hydrologic reasoning.

In extreme cases, the regressions can result in negative sensitivity index estimates, which are known to be unrealistic. We suggest setting those values to zero in applications.

In summary, the deduced functional relationships between basin characteristics and the sensitivity of major hydrologic processes shows promising results, indicated by strong coefficients of determination $R_{adj}^2$ between derived and predicted sensitivities. The large number of basins, which can span over multiple climatic regions, ensures transferability of the relationships to North American basins on which an xSSA sensitivity analysis has not explicitly been performed. The consistency of results with previous studies indicates their robustness with respect to other models (e.g., PRMS) and sensitivity metrics (e.g., FAST method based on mean streamflow).

## Discussion

This continental-scale sensitivity analysis of hydrologic model parameters and processes addresses the four key challenges raised in the introduction.

First, sensitivity analyses are traditionally carried out for model parameters only. This leads to difficulties in making high-level modeling decisions; these include, for example, whether to prioritize model development and improvement or to secure better datasets, which may reduce uncertainty. The novel analysis used herein extends the traditional Sobol' sensitivity analysis to cover model components or processes at the continental scale.

Second, due to the computationally expensive nature of sensitivity analyses, they are usually carried out at a small set of locations. This work successfully applies the novel sensitivity analysis over a large domain, allowing us to draw conclusions across multiple climatic regions. The analysis leads to hydrologically consistent and quantitative sensitivity index estimates—either aggregated in time or time-dependent—identifying quickflow as the overall most sensitive process in the Eastern United States, with Infiltration, Evaporation and Percolation processes being of secondary importance. In snow-dominated regions such as the Rocky Mountains, the Potential melt, Snow balance, Precipitation correction, and Rain-snow partitioning processes are of large importance. The time-resolved sensitivity of processes provides particularly detailed insights into the common sensitivity of processes across all watersheds. Furthermore, the large number of basins analyzed allows to deduce functional relationships between basin characteristics and climatic indices that allow estimating the sensitivity of processes, even without the need to perform the computationally expensive sensitivity analysis for basins not covered herein.

Third, sensitivity analyses are usually based on a single model, which limits the conclusions to certain model process definitions and presents the risk that sensitivity estimates might be different for other models. The use of the Raven hydrologic modeling framework[65] and the novel blended model structure of[47] allow a seamless analysis of a range of model structures, thereby reducing the risk of inferring model-specific conclusions as a range of model structures are analyzed simultaneously. It should be noted that the model structure can be inferred during model calibration based on observations, to build hydrological modeling hypotheses (e.g., McMillan et al.[66], McMillan et al.[67], Clark et al.[68], Fenicia et al.[69], and Fenicia et al.[70]). This work can help modelers decide on the model structure to use for inference of model structure in the following manner. First, processes with small sensitivities can be represented with essentially any option since the streamflow simulations are not affected in an important manner. Second, the processes with larger sensitivities should be included in the calibration process since the selection of the process option will play an important role in the streamflow simulations.

Fourth, sharing detailed results of large-scale model analyses is challenging due to the large amount and complexity of the data involved. The data and results used and produced in the work

presented herein are shared through an online mapping interface, which allows researchers to access and visualize inputs and results (see details under 'Data Availability'). This includes all calibration setups and results, the climatic indicators and basin properties, as well as individual sensitivity results and summary plots.

The xSSA sensitivity method can be applied to any kind of model. A comparison against the only other continental-scale parameter-based sensitivity analysis we are aware of shows good consistency with our results. It would be of interest to investigate if more distributed land-surface hydrologic models were to show similar patterns in these process sensitivities; and further, if the process sensitivities could then also be related to basin characteristics, as shown in this study. It would additionally be interesting to investigate the influence on the sensitivities when other forcing datasets are used or other modeling decisions—such as different numbers of soil layers, other process options, or different simulation time steps—are made. Our approach to continental-scale sensitivity analysis can be emulated for more complex land-surface hydrologic models, and the results suggest such an analysis could be conducted over a reduced number of watersheds. Such a framework could be readily extended to other modular environmental and ecological models, similarly challenged by high degrees of structural uncertainty.

This study is a quantitative continental sensitivity analysis of streamflow simulations to hydrologic processes- transferable both to other (i) basins and (ii) models using similar process representations.

## Methods

This work is out to analyze the sensitivity of streamflow to individual hydrologic processes across North America. This effort includes an attempt to deduce process sensitivities from basin characteristics alone without performing the expensive sensitivity analysis shown herein.

**Blended model structure.** The model used here is a blended model introduced by Mai et al.[47]. The unique definition of the blended model within the Raven hydrologic modeling framework[65] enables the seamless simulation of various model structures simultaneously. A process, for example Infiltration, is no longer defined by one specific process algorithm (e.g., the infiltration definition used in model A), but is now calculated as the weighted average flux from several independent algorithms (for example, infiltration algorithms used by models A, B and C). The specific process options and associated model parameters used here are available in Tables C1 and C2 of Mai et al.[47], and are added to the Supplementary Material of this publication for the convenience of the reader. The blended model has 35 model parameters and eight parameters describing the weights between 19 process options overall. In total, the following eleven hydrologic processes are distinguished: Infiltration $M$, Quickflow $N$, Evaporation $O$, Baseflow $P$, Snow Balance $Q$, Surface Runoff $R$, Delayed Runoff $S$, Potential Melt $T$, Percolation $U$, Rain-Snow Partitioning $V$, and Precipitation Correction $W$. The model setup used in this study is exactly the same as in Mai et al.[47], except that here, it is applied to more than 3000 basins. More details about the model, its parameters and processes can be found in the Supplementary Material as well as in Mai et al.[47].

**Basin database.** The overall sequence of experiments performed in this study can be found in Fig. 1. The analysis is based on the HYSETS dataset[48], comprised of 14,425 basins filtered by size and overall data availability and quality. Details on the basin properties can be found on the website associated with this publication, showing eight physiographic attributes[71] based on the void-filled HydroSHEDS digital elevation model (DEM) with a 30 arc-second resolution[72], eight landcover types[73] based on the 250 m North American Land Change Monitoring System (NACLMS) for 2010[74], and three climatic indicators[75] used by Knoben et al.[76] for each basin.

**Preliminary calibration and validation.** A preliminary calibration/validation experiment of the blended model is performed to demonstrate the basic adequacy of the recently introduced blended model, and to act as the basis for excluding poorly performing models from analysis. This basic calibration using a single evaluation metric will be the precursor to a more elaborate future multi-metric model calibration/validation study to be informed by the results found here.

The subset of 3826 basins with more than five years of observed streamflow data between January 1991 and December 2010 (calibration period) is calibrated using the Dynamically Dimensioned Search (DDS)[77] algorithm maximizing the Nash-Sutcliffe efficiency (NSE)[78]. A budget of 2000 model evaluations per basin and ten independent trials is used for DDS to calibrate the 35 model parameters and the 8 parameters defining the model structure. All model runs use a two-year warm-up

period (January 1989 to December 1990). Both calibration and sensitivity analysis results are found to be insensitive to warm-up period duration when this period is extended from 2 to 5 years (results not shown). The best performing parameter set of the ten trials is subsequently used for validation between January 1971 and December 1990. Only the 3005 basins with more than five years of observed streamflow available in this 20-year period are assessed in validation. The validation is performed to show that the overall performance during calibration is maintained during an independent time period. We assume that a strong validation performance indicates a low likelihood that the model parameters are overfitted during calibration. The general performance of the blended model in calibration and in validation compared to fixed model structures has been demonstrated in detail by Chlumsky et al.[79].

**Sensitivity analysis.** The 3316 basins with an NSE performance of 0.5 or higher during calibration are then analyzed regarding their simulated streamflow sensitivity. The method applied here is the extended Sobol' Sensitivity Analysis (xSSA) introduced by Mai et al.[47], and illustrated at one example catchment per climate zone. This work replicates the analysis for 3316 basins, with exactly the same parameter and analysis settings. For each basin, $K = 1000$ Sobol' reference sets were used, leading to a computational budget of $(43 + 2) \times K$ model runs to determine the sensitivities of the 43 parameters, $(27 + 2) \times K$ model runs to determine the sensitivities of the 19 process options as well as the eight parameters to weight these options, and $(11 + 2) \times K$ model runs to determine the overall sensitivity of the eleven processes. To assess the impact of a priori range specification for each parameter (Table S2 in Supplementary Material), we repeated the analysis for a subset of 150 basins using ranges for parameters $x_2$ to $x_6$ that were reduced by 20%.

The Sobol' main sensitivity index $S_i$, as well as the total sensitivity index $ST_i$, are determined for the parameters, process options, and processes. Both indices are time-dependent since they are derived regarding the simulated streamflow time series $Q(t)$. To aggregate the indices over time, the variance-weighted approach as introduced by Cuntz et al.[59] and used by Mai et al.[47] is implemented. These indices are denoted by $S_i^w = \overline{S_i(Q(t))}$ and $ST_i^w = \overline{ST_i(Q(t))}$, respectively. Note that the averaging is performed over the sensitivity indices rather than the streamflow time series as used by Markstrom et al.[34], where the sensitivity regarding, for example, the mean runoff $S_i(\overline{Q(t)})$, is estimated. This averaging order is expected to lead to major differences as models and sensitivities are known to be non-linear.

The xSSA analysis is performed using the blended model setup within the Raven modeling framework[65], but can also be performed with any modeling framework, such as SUMMA[80,81] and FUSE[82]. The process sensitivities can be estimated as soon as parameters are grouped into processes (no model implementation required), while properly analyzing multiple process options simultaneously requires the implementation of the weighted averages of these process option outputs (implementation in model most likely required).

**Deduction of relationships between basin attributes and sensitivity analysis results.** Finally, we attempt to infer functional relationships between known basin characteristics and the importance of these hydrologic processes using regressions. The twelve basin characteristics used as predictors are the logarithmic area of the basin in $[km^2]$, the mean elevation of the basin in $[m]$, the average slope of the basin in $[\circ]$, the fraction of the forest cover in $[km^2/km^2]$, the annual sum of precipitation in $[mm]$, the annual average temperature in $[\circ C]$, the annual sum of potential evapotranspiration in $[mm]$, the average annual number of days below $0\,°C$, the average annual number of days where precipitation is sleet, i.e., temperature between $-0.85\,°C$ and $1.15\,°C$, as well as the aridity $I_m$, seasonality $I_{m,r}$, and fraction of precipitation as snow $f_S$. The latter three are derived following Knoben et al.[76]. The regression analysis is performed using the Mathematica[83] function LinearModelFit by first fitting the 7 possible polynomials with up to degree three and up to five coefficients using each of the 12 combinations to pick one out of the twelve predictors. Second, we fit the 381 possible polynomials with up to degree three and up to six coefficients using each of the 66 combinations to pick two out of the twelve predictors. The regressions were performed in a cross-validation setup, i.e., each polynomial was fitted to two-thirds of the basins and validated using the remaining one-third of the basins. The polynomial with the best average adjusted coefficient of determination $R_{adj}^2$ in validation across 10 trials was identified- one best across all one-predictor polynomials and one best across the two-predictor polynomials. The latter two-predictor polynomial was chosen over the one-predictor version if the adjusted coefficient of determination was improved by at least 0.05. In total, $252,300\ (= 12 \times 7 \times 10 + 66 \times 381 \times 10)$ regressions were performed per process predicting the process sensitivity $ST_i^w$ of the nine processes. The two Baseflow and Convolution of delayed runoff processes are excluded as their weighted total Sobol' index $ST_i^w$ are smaller than 0.01 in all of the 3316 basins analyzed. The best regression function leading to the best adjusted coefficient of determination $R_{adj}^2$ is then used to perform a regression using all 3316 basins (reported in Table 1). To assess the robustness of this regression, we performed another cross-validation experiment similar to the previous one. The regression coefficients of the best regression (reported in Table 1) are refit now using two-thirds of the basins and validating them using the remaining one-third of the basins (reported in Fig. 5). This experiment is repeated with 100 random splits of basins.

## Data availability

The xSSA data generated in this study, including examples, can be found on GitHub (https://github.com/julemai/xSSA-North-America). They have been deposited in the Zenodo database under accession code https://doi.org/10.5281/zenodo.5730428. Additional information such as model setups and interactive visualization of data and results can be found on the webpage associated with this publication (http://www.hydrohub.org/sa_introduction.html#xssa-na) and in the Supplementary Material.

## Code availability

The xSSA code used to generate the results presented in this study, including examples, can be found on GitHub (https://github.com/julemai/xSSA-North-America). They have been deposited in the Zenodo database under accession code https://doi.org/10.5281/zenodo.5730428.

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

## Acknowledgements

This research was undertaken thanks in part to funding from the CANARIE research software funding program (project RS-332). The work was made possible by the facilities of the Shared Hierarchical Academic Research Computing Network (SHARCNET; www.sharcnet.ca) and Compute/Calcul Canada. The authors also thank the Canadian Foundation for Innovation John Evans Leaders Fund for the additional supercomputing support and resources. We thank Pawel Pomoski for the Dedicated Programming Support granted through the "SHARCNET Round XVI Programming Competition" to improve the performance of the xSSA code implementation and the addition of parallel computing capabilities.

## Author contributions

J.M. set up the models, implemented and performed model calibrations and sensitivity analyses, wrote main parts of the manuscript, prepared all figures and tables, created the website hosting all setups and results; J.R.C. contributed to the writing of the manuscript, implemented required modifications in Raven, provided feedback on model setups, helped to set up the model with the selected options and resolved inconsistencies in Raven detected by earlier versions of the sensitivity analysis, and helped with the hydrologic interpretation of the results; B.A.T. contributed to the experimental design, the writing of the manuscript, provided feedback on the manuscript and helped with the hydrologic interpretation of the results; R.A. provided meteorologic forcings and geophysical data to set up the watersheds used in this study and helped with the hydrologic interpretation of the results.

## Competing interests

The authors declare no competing interests.
