## [Peer Review File · Nature Communications]

Reviewers' Comments:

Reviewer #1:

Remarks to the Author:

I really enjoyed reading and reviewing the manuscript entitled "The Sensitivity of Simulated Streamflow to Individual Hydrologic Processes Across North America". I have found that this paper would be a great addition to the field of hydrological modeling and its sensitivity analyses. I especially like the broader applicability (i.e., continental, and subcontinental domain) of the proposed methodology and the demonstration of a lot of results (figures) in a very informative and appropriate manner. However, there are some sections/spaces of improvement. Please find my questions and detailed review in the following section.

Major comments:

1. Authors did not highlight major/important findings in the abstract of the paper. Authors should highlight some main outcomes from the study, in one line, in the abstract.
2. How did the authors decide warmup time period?
3. The model warmup is only for two years (1989-1990), did the authors consider checking the model equilibrium states for the water balance components?
4. Does the model come in equilibrium only after two years, as I suspect, many of the previous studies show that various models take more than five years to come to equilibrium?
5. Line 249: Authors should provide more details about the "Blended Model Structure" in the supplementary document.

Minor comments:

6. Line 11: 'the estimation'- incorrect article.
7. Line 10 to 13: long sentence, consider breaking it into two.
8. Line 13 to 15: confusing sentence not clear and inappropriate articles used in the sentence.
9. Line 232 to 235: did the authors provide a link for the online mapping interface?
10. Line 237 to 239: confusing sentence, consider rewriting it.
11. Line 306 to 308: long sentence, consider breaking it into two.
12. At many instances throughout the paper, there were frequent grammatical mistakes (articles and verbs). Consider checking the whole manuscript carefully.
13. Line 312: Data availability link does not work. Will it be available after publication?

Reviewer #2:

Remarks to the Author:

Summary

In this paper, the authors present the results from a novel approach to quantify the sensitivity of streamflow simulations to hydrological processes across a large sample of catchments (>3,000) in North America. The framework builds upon the Raven hydrological modeling platform (Craig et al. 2020), which includes several options for a suite of process modeling decisions, and has the unique feature that each hydrological variable (e.g., infiltration) is computed from the weighted average of all model options. This "blended model" is first calibrated and validated in all catchments by maximizing the Nash-Sutcliffe efficiency (NSE; Nash and Sutcliffe 1970) for daily flows. The catchments that provide NSE > 0.5 are used for subsequent analyses, which consists on applying the extended Sobol' sensitivity analysis method (Mai et al. 2020) to quantify hydrologic process sensitivities. The authors characterize spatial patterns of sensitivity indices associated to different processes (e.g., quickflow, snow balance, evaporation, potential melt, etc.), and provide detailed characterizations of sensitivities for eight basins representative of different climatic regions. Finally, the authors demonstrate the potential of this new database to predict process sensitivities as a function of catchment climatic and physiographic descriptors.

The paper is well organized, well written, and the results are nicely presented. I commend the authors for the quality of the graphics and for setting up a really nice website with all the results obtained from their analyses. Despite the massive effort on including a large number of basins and model simulations, there are some modeling decisions that, in my opinion, may affect the catchment selection process and the realism of process sensitivities, setting this study more as a methodological demonstration rather than a contribution towards improved hydrologic process understanding. Hence, I think the authors should address three main observations before this manuscript is considered for publication. There are other minor comments and editorial suggestions that may also help the authors to improve the quality of their manuscript.

Major comments

1. Model calibration/validation: In my opinion, there is a fundamental contradiction between using a modular modeling framework like Raven, which should motivate careful hypothesis testing of hydrological models using a diagnostic model evaluation framework (with multiple data sources), and the calibration approach adopted here, based solely on the NSE, whose weaknesses have been discussed during more than one decade (e.g., Schaefli and Gupta 2007; Gupta et al. 2009 and many others). Further, it is likely that some process options in specific catchments are given unrealistic weights for the selected NSE threshold (0.5). I recommend the authors to refine their basin selection increasing the value of such threshold (perhaps 0.7; please check Moriasi et al. 2007), using additional process-oriented performance metrics (e.g., Pfannerstill et al. 2014; Addor et al. 2018) and, at the very least, look at the time series of fluxes and states obtained with the blended model, and their seasonal cycles across all catchments to ensure that model outputs make sense (e.g., check that there are not unrealistic trends in soil moisture, snow towers, etc.).

2. Sensitivity analysis: My main concern is that the selected parameter bounds may be inappropriate (i.e., too wide) for a large fraction of catchments, providing unrealistic sensitivity indices for some processes. Did the authors use the parameters and weights obtained from the calibration process to inform the parameter ranges used in the xSSA method? Or did the authors examine the sensitivity of the results (and conclusions) to the choice of parameter bounds, at least for the catchments in Figure 4? If not, I suggest doing this exercise to ensure that the conclusions presented in this paper are well supported.

3. Statistical model: I think that there are several methodological decisions that need justification and/or further explanation, including:

- What is the rationale of considering latitude and longitude among the pool of potential predictors? Please provide a reasonable justification, since they do not necessarily explain the spatial variability of catchment responses, which result from the combination of climatic and physiographic characteristics that can be highly heterogeneous in relatively short distances.
- Did you start with the simple exercise of plotting your sensitivity indices against your catchment descriptors (i.e., x , xy , y^2 , etc.), and computing correlation coefficients (both Pearson and Spearman) and p-values, in order to justify the statistical model structure? Did you check the multi-collinearity among predictors?
- Finally, if the goal is to build statistical models for prediction, a cross-validation type metric (like GCV or PRESS) would have been a better choice to select the best model. Additionally, the authors should have used the adjusted R^2 to compare models with different numbers of predictors (instead of the coefficient of determination, R^2).

Specific comments

4. L8-9: I would be careful with this statement, since the sensitivity of any hydrological variable can be quantified, more generally, by perturbing factors, which may also include input forcing variables

such as precipitation or temperature (e.g., Raleigh et al. 2015).

5. L16: From my understanding of the paper, the estimation of 'entire process sensitivities' includes parameters (right?). If yes, I think it would be good to clarify here.

6. L21: I don't think that reference [2] is appropriate in the context of hydrological models for flood forecasting (indeed, Cao and Zhang do not mention hydrological models at all), and the authors could cite more relevant papers in this field (e.g., Weerts and El Serafy 2006; Todini 2008; Bogner and Pappenberger 2011; Rakovec et al. 2015).

7. L23-24: I agree, but the authors should also recognize that a thorough model evaluation that takes advantage of information theory and newly available datasets is critical to advance hydrologic process understanding, and this is a good place to mention this, with proper citations (e.g., Gupta et al. 2008; Hrachowitz et al. 2014; Nijzink et al. 2018; Khatami et al. 2019; Széles et al. 2020; Dembélé et al. 2020).

8. L26-28: in a broader sense, SA is used to identify the most sensitive factors, which could be more than just model parameters (see comment #4). It is also confusing that the authors refer to 'parametric' models, since the paper is about process-based hydrological modeling (which contain many parameters).

9. L28: I'm not sure that Konapala et al. (2020) should be listed here, since they did not conduct formal sensitivity analyses.

10. L29: 'SA is a well-established method'. Perhaps it would be more appropriate using the word 'tool', since there are many SA methods.

11. Figure 1: it is common practice to start hydrological analyses (especially with numerical, process-based models) at the beginning of the water year, when catchments reach the 'minimum' water storage. Please provide an explanation of why you decided to start on January 1st.

12. L79-81: does 'in agreement with expectations' only refer to snow balance, or in general to all processes?

13. L102: 'The analyses presented here... provide insights over several model structures'. I would delete statements like this from the manuscript, since the results do not support clear recommendations on which model structures (i.e., dominant processes, process parameterizations, model architecture, spatial connectivity, etc.) are more appropriate in different regions.

14. L114: please provide a reference for c-means fuzzy clustering.

15. L148: the dark green for baseflow is really hard to see.

16. L148-149: I do not see quick flow as 'almost constant' in Figure 4f. Did you look at other basins in the same cluster?

17. L166: I suggest changing the Greek letter currently being used by 'r', since rho is normally reserved for the Spearman rank correlation coefficient.

18. L166: I don't think that 'reliability' is the right word for this context, since it typically refers to an attribute of probabilistic simulations or predictions. I would recommend using 'stability' or a similar word.

19. L168: Please clarify if you used all 3,316 catchments to derive the coefficients in Table 1. What

period did you use to compute climate attributes?

20. L197-198: It is really hard to believe that altitude does not play a considerable role here, but longitude does (see major comment #3).

21. L225: do you actually need thousands of basins for deriving these 'functional relationships'?

22. L231: 'reducing the risk of inferring model-specific conclusions'. I disagree with this statement, since such 'risk' should be minor if you select your model based on careful model structure diagnostics using field observations and in situ measurements. Additionally, the concept of 'blended model' does not really help to reduce structural and parametric uncertainties if model evaluation is simply conducted by NSE or KGE. I think the authors should recognize this limitation, and also acknowledge a large body of work based on inference from observations to build hypotheses in hydrological modeling (e.g., McMillan et al. 2011, 2014; Clark et al. 2011; Fenicia et al. 2014, 2016) and work towards 'the right answers for the right reasons' (Kirchner 2006).

23. L234: although I commend the authors for the fantastic outcome (a very nice website with a lot of information), I think it would tremendously benefit from including more graphics (time series with simulated and observed streamflow, scatterplots, simulated and observed flow duration curves, seasonal runoff, etc.) of calibration and validation model results.

24. L256: 'eight parameters'. I assume that you are referring to r_1 , r_2 , etc., which define the weights in Raven. Perhaps it would be better referring to them as 'coefficients' instead.

25. L259: I think that the manuscript should be self-contained and, at the very least, a better description and list of parameters (including and justifying the choice of ranges) should be included. Did you implement the models in lumped mode, as Mai et al. (2020) and Chlumsky et al. (2021)? Did you calibrate 35 + 8 parameters in all catchments?

26. L271-272: Did you find large differences between parameter sets and NSEs from the 10 trials, to justify doing this for more than 3,000 catchments?'

27. L280: 'demonstrated at one example catchment per climate zone'. This is confusing. I understand that you applied the method for all catchments, and NOT a few example basins. Perhaps a better word would be "illustrated".

28. References: I'm not sure that the format is correct, since it is not possible to see the title of the papers.

Suggested edits

29. L11: 'based on basins' reads awkward. Perhaps it would be better using 'based on catchments'.

30. L11: 'enabling the estimation of the process sensitivities' -> 'to estimate process sensitivities'.

31. L12: 'needing' -> 'the need'.

32. L13: delete 'itself'.

33. L13: delete continuously.

34. L32 and everywhere else: I suggest using 'indices' instead of 'indexes'.

35. L32: 'hydrologic cycle' -> 'water cycle'.

36. L47: 'structure' -> 'structures'.

37. Caption of Figure 1: 'The reasons for reducing the sets of basins from the original 14 425 basins in the HYSETS database to 3316 basins'. I would delete the third 'basins' (and perhaps also the second).

38. L67: delete 'here used'.

39. L69: 'are not considered' -> 'are not included'.

40. L128-150: I recommend writing the full names for all months.

41. L174: 'the ones that are' -> 'those'.
42. L180: insert comma after 'this process'.
43. L185-186: I suggest placing the following text between hyphens or commas: 'already identified to be important in the Rocky Mountains and the higher latitudes starting over the Great Lakes (see Fig. 3)'.
44. L200: 'wetness and presence of snow are both key indicators' -> 'both wetness and presence of snow are key indicators'.
45. L201: 'a best predictor' -> 'the best predictor'.
46. L202: insert comma after 'in extreme cases'.
47. L208: 'indicates robustness of the results and the transferability' -> 'indicates their robustness'. I suggest deleting 'transferability'.
48. L211: 'This large-scale continental sensitivity analysis' -> 'This continental-scale sensitivity analysis'.
49. L222: insert comma after 'Mountains'.
50. L225: 'analyses' -> 'analyzed'.
51. L242-243: 'approximation of the sensitivity of hydrologic processes on streamflow simulations'. I suggest re-writing as 'sensitivity of streamflow simulations to hydrologic processes'.

References

- Addor, N., G. Nearing, C. Prieto, A. J. Newman, N. Le Vine, and M. P. Clark, 2018: A Ranking of Hydrological Signatures Based on Their Predictability in Space. *Water Resour. Res.*, 54, 8792–8812, doi:10.1029/2018WR022606.
- Bogner, K., and F. Pappenberger, 2011: Multiscale error analysis, correction, and predictive uncertainty estimation in a flood forecasting system. *Water Resour. Res.*, 47, W07524, doi:10.1029/2010WR009137.
- Chlumsky, R., J. Mai, J. R. Craig, and B. A. Tolson, 2021: Simultaneous Calibration of Hydrologic Model Structure and Parameters Using a Blended Model. *Water Resour. Res.*, 57, 1–22, doi:10.1029/2020WR029229.
- Clark, M. P., H. K. McMillan, D. B. G. Collins, D. Kavetski, and R. A. Woods, 2011: Hydrological field data from a modeller's perspective: Part 2: process-based evaluation of model hypotheses. *Hydrol. Process.*, 25, 523–543, doi:10.1002/hyp.7902.
- Craig, J. R., and Coauthors, 2020: Flexible watershed simulation with the Raven hydrological modelling framework. *Environ. Model. Softw.*, 129, 104728, doi:10.1016/j.envsoft.2020.104728. <https://doi.org/10.1016/j.envsoft.2020.104728>.
- D. N. Moriasi, J. G. Arnold, M. W. Van Liew, R. L. Bingner, R. D. Harmel, and T. L. Veith, 2007: Model Evaluation Guidelines for Systematic Quantification of Accuracy in Watershed Simulations. *Trans. ASABE*, doi:10.13031/2013.23153.
- Dembélé, M., N. Ceperley, S. J. Zwart, E. Salvatore, G. Mariethoz, and B. Schaefli, 2020: Potential of satellite and reanalysis evaporation datasets for hydrological modelling under various model calibration strategies. *Adv. Water Resour.*, 143, doi:10.1016/j.advwatres.2020.103667.
- Fenicia, F., D. Kavetski, H. H. G. Savenije, M. P. Clark, G. Schoups, L. Pfister, and J. Freer, 2014: Catchment properties, function, and conceptual model representation: Is there a correspondence? *Hydrol. Process.*, 28, 2451–2467, doi:10.1002/hyp.9726.
- , —, H. H. G. Savenije, and L. Pfister, 2016: From spatially variable streamflow to distributed hydrological models: Analysis of key modeling decisions. *Water Resour. Res.*, 52, 954–989, doi:10.1002/2015WR017398.
- Gupta, H. V., T. Wagener, and Y. Liu, 2008: Reconciling theory with observations: elements of a diagnostic approach to model evaluation. *Hydrol. Process.*, 22, 3802–3813, doi:10.1002/hyp.
- , H. Kling, K. K. Yilmaz, and G. F. Martinez, 2009: Decomposition of the mean squared error and NSE performance criteria: Implications for improving hydrological modelling. *J. Hydrol.*, 377, 80–91, doi:10.1016/j.jhydrol.2009.08.003.
- Hrachowitz, M., and Coauthors, 2014: Process consistency in models: The importance of system

signatures, expert knowledge, and process complexity. *Water Resour. Res.*, 50, 7445–7469, doi:10.1002/2014WR015484.

Khatami, S., M. C. Peel, T. J. Peterson, and A. W. Western, 2019: Equifinality and Flux Mapping: A New Approach to Model Evaluation and Process Representation Under Uncertainty. *Water Resour. Res.*, 55, 8922–8941, doi:10.1029/2018WR023750.

Kirchner, J. W., 2006: Getting the right answers for the right reasons: Linking measurements, analyses, and models to advance the science of hydrology. *Water Resour. Res.*, 42, W03S04, doi:10.1029/2005WR004362.

Konapala, G., S. C. Kao, and N. Addor, 2020: Exploring Hydrologic Model Process Connectivity at the Continental Scale Through an Information Theory Approach. *Water Resour. Res.*, 56, doi:10.1029/2020WR027340.

Mai, J., J. R. Craig, and B. A. Tolson, 2020: Simultaneously determining global sensitivities of model parameters and model structure. *Hydrol. Earth Syst. Sci.*, 24, 5835–5858, doi:10.5194/hess-24-5835-2020.

McMillan, H., M. Gueguen, E. Grimon, R. Woods, M. Clark, and D. E. Rupp, 2014: Spatial variability of hydrological processes and model structure diagnostics in a 50 km² catchment. *Hydrol. Process.*, 28, 4896–4913, doi:10.1002/hyp.9988.

McMillan, H. K., M. P. Clark, W. B. Bowden, M. Duncan, and R. A. Woods, 2011: Hydrological field data from a modeller's perspective: Part 1. Diagnostic tests for model structure. *Hydrol. Process.*, 25, 511–522, doi:10.1002/hyp.7841.

Nash, J., and J. Sutcliffe, 1970: River flow forecasting through conceptual models part I - A discussion of principles. *J. Hydrol.*, 10, 282–290, doi:10.1016/0022-1694(70)90255-6.

Nijzink, R. C., and Coauthors, 2018: Constraining Conceptual Hydrological Models With Multiple Information Sources. *Water Resour. Res.*, 54, 8332–8362, doi:10.1029/2017WR021895.

Pfannerstill, M., B. Guse, and N. Fohrer, 2014: Smart low flow signature metrics for an improved overall performance evaluation of hydrological models. *J. Hydrol.*, 510, 447–458, doi:10.1016/j.jhydrol.2013.12.044.

Rakovec, O., A. H. Weerts, J. Sumihar, and R. Uijlenhoet, 2015: Operational aspects of asynchronous filtering for flood forecasting. *Hydrol. Earth Syst. Sci.*, 19, 2911–2924, doi:10.5194/hess-19-2911-2015.

Raleigh, M. S., J. D. Lundquist, and M. P. Clark, 2015: Exploring the impact of forcing error characteristics on physically based snow simulations within a global sensitivity analysis framework. *Hydrol. Earth Syst. Sci.*, 19, 3153–3179, doi:10.5194/hess-19-3153-2015.

Schaefli, B., and H. V Gupta, 2007: Do Nash values have value? *Hydrol. Process.*, 21, 2075–2080, doi:10.1002/hyp.

Széles, B., J. Parajka, P. Hogan, R. Silasari, L. Pavlin, P. Strauss, and G. Blöschl, 2020: The Added Value of Different Data Types for Calibrating and Testing a Hydrologic Model in a Small Catchment. *Water Resour. Res.*, 56, doi:10.1029/2019WR026153.

Todini, E., 2008: A model conditional processor to assess predictive uncertainty in flood forecasting. *Int. J. River Basin Manag.*, 6, 123–137, doi:10.1080/15715124.2008.9635342.

Weerts, A. H., and G. Y. H. El Serafy, 2006: Particle filtering and ensemble Kalman filtering for state updating with hydrological conceptual rainfall-runoff models. *Water Resour. Res.*, 42, 1–17, doi:10.1029/2005WR004093.

Reviewer #3:

Remarks to the Author:

In this study, the authors conducted a very comprehensive hydrologic model sensitivity analysis using the Raven hydrologic modeling framework for over 3000 basins across the North America. The authors rigorously calibrated, validated, and selected qualified basins to support their analysis, and then concisely reported their sensitivity analysis findings by major hydrologic processes in space and time. The manuscript was nicely written, clearly organized, and provided useful scientific insights to the broader hydrologic modeling community for the development and enhancement of future hydrologic

models. Based on what I know, this is by far the most elaborated and comprehensive hydrologic model sensitivity analysis in the North America, and hence provides timely importance for a highly impactful journal like Nature Communications. I would gladly recommend accepting this manuscript, after the following concerns have been satisfactorily addressed in the revised manuscript.

1. While I agree with the flexibility of Raven, it is still a form of model-specific, multi-structural hydrologic models (like FUSE and SUMMA). It does provide alternative hydrologic processes for different location-specific needs, but is also unique to its non-structural features such as software workflow, numerical solvers, and spatial-temporal meshing strategies. In addition, unless you considered the uncertainties associated with non-calibrated land surface parameters and meteorologic forcing datasets, your findings should also be specific to your applications. In my opinion, the authors should provide additional caveats and discuss the potentially different findings when using other multi-structural hydrologic models, land surface parameter sets, meteorologic observations, and hydrologic modeling strategies.

2. There seems to be a large number of model parameters in Raven (35 model parameters and 8 process combinations). How can one ensure that we have sufficiently long historical observations for calibration? Have you also evaluated other non-streamflow observations such as snow or evapotranspiration? In other words, how can one ensure that this is not an underdetermined question? Please comment on this challenge in the revised manuscript.

3. It is interesting to see that the authors found unclear sensitivities associated with the convolution processes which control the timing of runoff. I am wondering to what level this may be affected by the timing issues within the meteorologic observations? To be more specific, a recent study by Pierce et al. (2021) suggested that the timing adjustment used in the Livneh et al. (2015) dataset (used in HYSETS) may be unsuitable and can lead to underestimation of hydrologic extremes. This issue was caused by the varying reporting time of daily rainfall stations, which usually reported daily rainfall from 7am to 7am instead of midnight to midnight. In other words, how was the rainfall timing issue addressed in HYSETS? Do the meteorologic observations in HYSETS have sufficiently accurate timing to help you evaluate the sensitivity of hydrologic processes associated with runoff timing?

4. The study built on Chlumsky et al. (2021), which reported detailed calibration findings of 12 MOPEX12 catchments. Since the number of basins has been largely expanded in this study (to over 3000), I think the readers would also like to see some further information from this more comprehensive calibration. For instance, among the alternative hydrologic processes, which ones typically have the largest weights across over 3000 basins?

Reference

Pierce et al. (2021), An Extreme-preserving Long-term Gridded Daily Precipitation Data Set for the Conterminous United States, Journal of Hydrometeorology, <https://doi.org/10.1175/JHM-D-20-0212.1>

Reviewers' comments:

Reviewer #1:

I really enjoyed reading and reviewing the manuscript entitled "The Sensitivity of Simulated Streamflow to Individual Hydrologic Processes Across North America". I have found that this paper would be a great addition to the field of hydrological modeling and its sensitivity analyses. I especially like the broader applicability (i.e., continental, and subcontinental domain) of the proposed methodology and the demonstration of a lot of results (figures) in a very informative and appropriate manner. However, there are some sections/spaces of improvement. Please find my questions and detailed review in the following section.

We are glad that the reviewer enjoyed reading our manuscript; that's what every author wants to read when getting back the reviews. Thanks for this.

We highly appreciate the reviewer's time and efforts in evaluating our manuscript and provide a detailed response below. We hope that the additional analysis of the warm-up periods and the additional information of the "Blended Model Structure" now provided in the Supplements are helpful. Thanks for helping us improving our manuscript!

Major comments:

1. *Authors did not highlight major/important findings in the abstract of the paper. Authors should highlight some main outcomes from the study, in one line, in the abstract.*

We wish we could be more elaborate in the abstract and extend it to include results. The limit of 150 words however makes this very difficult. We completely revised the abstract to address this comment and comments raised by the other reviewers. We focused on describing what we did, why this analysis is important and the high-level conclusions that can be drawn using this analysis. Major outcomes now highlighted are the importance of quickflow across North America and the dissemination of data and results through a website.

2. *How did the authors decide warmup time period?*

The warmup period was chosen based on data availability and by checking for initial state equilibrium of the simulated discharge and additional model states for several, randomly selected basins.

3. *The model warmup is only for two years (1989-1990), did the authors consider checking the model equilibrium states for the water balance components?*

We did this for several basins (as mentioned above) when we decided on the experimental setup. We indeed increased the warmup period from initially one year to two years during that testing phase.

4. *Does the model come in equilibrium only after two years, as I suspect, many of the previous studies show that various models take more than five years to come to equilibrium?*

We agree that warmup period lengths might be model dependent on the basins but also the parameter sets chosen for simulation. We wish to emphasize that this is much more important when the model's performance is evaluated (in, for example, calibration experiments) and of lower importance during sensitivity analysis when the sensitivity is not based on observations. The sensitivity is purely derived from simulations. The quality of those individual simulations is not evaluated (nor is this generally done in global sensitivity analyses).

In response to this comment, we re-derived the model performances using the

Figure R1: **Impact of the warm-up period on calibration results.** The left panel shows a 2D histogram of the performance with two years warmup (as used in this study) and a five-years warmup period. The models are not recalibrated. The performances using five years of warmup are based on the calibrated model using two years of warmup. The right panel shows the CDF of the model performances of the 3826 basins that are at least 300 km² in size and have at least five years of streamflow observations between 1991 and 2010.

first five years of the simulation (1989-1993) as warmup and the remaining 17 years (1994-2010) to derive the model performance. The results are shown in Fig. R1 and are a clear demonstration that extending the warmup period has only minor effects on the performance of the model regarding streamflow simulations.

The same approach was applied to recalculate the sensitivities of the processes based on the time period from 1994 to 2010 leaving now five years (1989 to 1993) for warmup. The results shown in Fig. R2 demonstrate that the sensitivities are very stable with mean absolute differences (MAD) of no more than 0.005. The correlation coefficients r of the process sensitivities ST_i^w comparing two years and five years of warmup are all above 0.995.

We want to note that the differences between the two estimates can be either caused by the longer warmup period or the fact that the sensitivity estimates are now based on a different sample of meteorological forcings.

We hope the reviewer agrees that these results emphasize the minor impact of the warmup period on the sensitivity results and that the choice of a longer warmup would not change any conclusion of this work. As such, the only change to the manuscript for this comment is as follows:

line 307 ff. Both calibration and sensitivity analysis results are found to be insensitive to warm-up period duration when this period is extended from 2 to 5 years (results not shown).

5. *Line 249: Authors should provide more details about the “Blended Model Structure” in the supplementary document.*

Thanks for pointing this out. We agree and added the section “Details about the Blended Model” to the Supplementary Material. Most of the content (parameter

Figure R2: **Impact of warm-up period on xSSA results.** The 2D histograms show the process sensitivities ST_i^w derived with the original two-year warmup used throughout the presented study (x-axis) and compare them to the sensitivities derived based on the time period 1994 to 2010 while assigning the first five years of the simulation as warmup (1989-1993) (y-axis). The results are shown for all the 3316 basins of the study as this did not require new model runs only the derivation of the sensitivities of a shorter time period. The model results (i.e. streamflow simulations) of the original runs covering 1991-2010 were saved. The correlation coefficients r and mean absolute differences MAD between the two estimates are added as labels to each panel. The results of the two least sensitive processes ‘Baseflow’ and ‘Convolution (delayed runoff)’ are not shown as all sensitivities are below 0.01 in each basin.

tables, flowcharts, etc) are the same as in the publication introducing the Blended Model [Mai et al., 2020]. We mention those additional resources now in the manuscript:

line 282 ff. The specific process options and associated model parameters used here are available in Tables C1 and C2 of Mai et al. [2020], and are added to the Supplementary Material of this publication for the convenience of the reader.

line 289 ff. More details about the model, its parameters and processes can be found in the Supplementary Material as well as in Mai et al. [2020].

Minor comments:

6. *Line 11: ‘the estimation’- incorrect article.*

We modified the sentence entirely. It now reads as follows:

line 10 ff. In this study, we apply a novel analysis over more than 3000 basins across North America considering a blended model structure, which includes not only parametric, but also structural uncertainties.

7. *Line 10 to 13: long sentence, consider breaking it into two.*

Agreed. We rearranged the abstract entirely and hope that it contains shorter sentences and more results.

8. *Line 13 to 15: confusing sentence not clear and inappropriate articles used in the sentence.*

We adjusted the sentence as follows and hope it is more clear now:

line 13 ff. It also leads to high-level conclusions about the importance of water cycle components on streamflow predictions, such as quickflow being the most sensitive process for streamflow simulations across the North American continent.

9. *Line 232 to 235: did the authors provide a link for the online mapping interface?*

Yes, the website was already accessible for the initial submission (Link in ‘Data Availability section’; see also your comment #13). It should be accessible via: <http://www.civil.uwaterloo.ca/xSSA/webpage/index.html>. We also mention the Data Availability section in the manuscript now.

line 257 ff. The data and results used and produced in the work presented herein are shared through an online mapping interface, which allows researchers to access and visualize inputs and results (see details under ‘Data Availability’).

10. *Line 237 to 239: confusing sentence, consider rewriting it.*

We made the following adjustments and hope the statement appears more clear now:

line 262 ff. It would be of interest to investigate if more distributed land-surface hydrologic models were to show similar patterns in these process sensitivities; and further, if the process sensitivities could then also be related to basin characteristics, as shown in this study.

11. *Line 306 to 308: long sentence, consider breaking it into two.*

We apologize for this and rewrote the sentence (now three) as follows:

line 359 ff. The regression function leading to the best adjusted coefficient of determination R_{adj}^2 is then used to perform a regression using all 3316 basins (reported in Table I). To assess the robustness of this regression, we performed another cross-validation experiment similar to the previous one. The regression coefficients of the best regression (reported in Table I) are refit now using two-thirds of the basins and validating them using the remaining one-third of the basins (reported in Fig. 5).

12. *At many instances throughout the paper, there were frequent grammatical mistakes (articles and verbs). Consider checking the whole manuscript carefully.*

Thanks for this comment. We decided to send the manuscript to a professional proof-reading service and hope this resolved the remaining grammatical mistakes.

13. *Line 312: Data availability link does not work. Will it be available after publication?*

The website was already accessible at the time of the initial submission. We regret that the reviewer was not able to access it and assume that this was because the URL included a line break that led to a broken link when clicking on it. It should be accessible via:

<http://www.civil.uwaterloo.ca/xSSA/webpage/index.html>

Reviewer #2:

Summary

In this paper, the authors present the results from a novel approach to quantify the sensitivity of streamflow simulations to hydrological processes across a large sample of catchments (>3,000) in North America. The framework builds upon the Raven hydrological modeling platform (Craig et al. [2020]), which includes several options for a suite of process modeling decisions, and has the unique feature that each hydrological variable (e.g., infiltration) is computed from the weighted average of all model options. This “blended model” is first calibrated and validated in all catchments by maximizing the Nash-Sutcliffe efficiency (NSE; Nash and Sutcliffe [1970]) for daily flows. The catchments that provide $NSE > 0.5$ are used for subsequent analyses, which consists on applying the extended Sobol’ sensitivity analysis method (Mai et al. [2020]) to quantify hydrologic process sensitivities. The authors characterize spatial patterns of sensitivity indices associated to different processes (e.g., quickflow, snow balance, evaporation, potential melt, etc.), and provide detailed characterizations of sensitivities for eight basins representative of different climatic regions. Finally, the authors demonstrate the potential of this new database to predict process sensitivities as a function of catchment climatic and physiographic descriptors.

We would like to first and foremost thank the reviewer for the thorough evaluation of our manuscript and the great suggestions helping to improve our manuscript and rule out ambiguities of our previous manuscript. Thanks a lot for your time and efforts!

The paper is well organized, well written, and the results are nicely presented. I commend the authors for the quality of the graphics and for setting up a really nice website with all the results obtained from their analyses. Despite the massive effort on including a large number of basins and model simulations, there are some modeling decisions that, in my opinion, may affect the catchment selection process and the realism of process sensitivities, setting this study more as a methodological demonstration rather than a contribution towards improved hydrologic process understanding. Hence, I think the authors should address three main observations before this manuscript is considered for publication. There are other minor comments and editorial suggestions that may also help the authors to improve the quality of their manuscript.

Thanks for your kind words. We are happy that the reviewer liked the analysis and the website accompanying this work. We reply in detail to all the comments and concerns raised below and hope that the reviewer agrees with the reply to the three major concerns:

- 1. The model ‘calibration’ is used to rule out the possibility that we base our final conclusions on a model that might not be appropriate. The calibration results are intended only to exclude from the analysis clearly poorly performing models, the calibration results/models are not otherwise used. This is admittedly not standard practice in sensitivity analyses (though perhaps should be!). We intend to use the preliminary calibration results of this work to inform our follow up study of a continental calibration of this model.**
- 2. We hopefully can convince the reviewer regarding the chosen ranges of the parameters and the minor impact a change of parameter ranges has on the conclusions drawn based on this work. Again, parameter ranges are not known a priori (as for any other hydrologic model) and their choice is based on expert knowledge. Model ranges could feasibly be reduced using calibration experiments but we perform this study to inform our calibration and further model development: sensitivity analysis is typically the precursor to detailed calibration, rather than the other way around. However, we performed additional experiments to demonstrate that the reduction of ranges (based on our preliminary ‘calibration’ experiments) has little effect upon the results.**

3. We revised the statistical model choice and evaluation as suggested by the reviewer but, again, those changes have little impact on the final results. The conclusions of this study are unchanged. But we absolutely agree that it is now more statistically sound. We appreciate the suggestions.

Major comments

1. *Model calibration/validation: In my opinion, there is a fundamental contradiction between using a modular modeling framework like Raven, which should motivate careful hypothesis testing of hydrological models using a diagnostic model evaluation framework (with multiple data sources), and the calibration approach adopted here, based solely on the NSE, whose weaknesses have been discussed during more than one decade (e.g., Schaeftli and Gupta [2007]; Gupta et al. [2009] and many others). Further, it is likely that some process options in specific catchments are given unrealistic weights for the selected NSE threshold (0.5). I recommend the authors to refine their basin selection increasing the value of such threshold (perhaps 0.7; please check Moriasi et al. [2007]), using additional process-oriented performance metrics (e.g., Pfannerstill et al. [2014]; Addor et al. [2018]) and, at the very least, look at the time series of fluxes and states obtained with the blended model, and their seasonal cycles across all catchments to ensure that model outputs make sense (e.g., check that there are not unrealistic trends in soil moisture, snow towers, etc.).*

We apologize again for the confusion we might have caused with presenting “calibration” results in this study. We are aware that this is usually not done in sensitivity analyses as usually the results of a SA are used to inform model improvement and calibration rather than the other way around. The calibration doesn’t inform the SA other than (1) to demonstrate that the blended model performs capably across most of the sites and (2) to remove clearly incapable models/basins from the analysis. This pre-selection and evaluation step is not standard in sensitivity analysis studies, but we thought that it would improve trust in the model and therefore the SA results. Future studies will be conducted to analyse the fluxes and states of models as well as setting up proper calibration experiments informed by the analysis presented here, i.e. selecting only sensitive parameters and only looking at states/fluxes that are sensitive when calibrated against streamflow.

To address your concerns with respect to the NSE threshold used, we repeated the analysis in deriving the functional relationships based on the 2243 basins that yield NSE performances of 0.7 and above during calibration. The results are shown in Fig. R4 yielding very similar results than the results presented in our initial submission (compare to original Fig. 5). Also, the predictors and functional relationships derived are basically the same as before. The reason for this is that the performance of a model in a given basin is not correlated to its process sensitivity. The model performance is derived based on comparisons with streamflow observations but these observations are not used to derive sensitivities. Due to the similarity of results using NSE thresholds of 0.5 and 0.7, we decided to continue to report all statistics and analyses based on the full set of basins with NSE performances above 0.5.

We apologize for the confusion the term ‘calibration/validation’ might have initially caused and use the terms ‘preliminary calibration/validation’ now throughout the manuscript hoping that this makes the sequence of experiments and future studies more clear. A few examples are the following:

line 298 ff. Preliminary Calibration and Validation

line 299 ff. A preliminary calibration/validation experiment of the blended model is performed to demonstrate the basic adequacy of the recently introduced

blended model, and to act as the basis for excluding poorly performing models from analysis. This basic calibration using a single evaluation metric will be the precursor to a more elaborate multi-metric model calibration/validation study to be informed by the results found here.

line 78 ff. It is important to note that the results of this calibration exercise are used to (1) exclude clearly low quality models from further analysis and (2) to demonstrate the basic adequacy of the models for simulating streamflow over the range of simulated conditions. While a more elaborate calibration study may improve individual optimal model performances, it is unlikely to yield improved global sensitivity estimates. The exclusion of low quality models is admittedly not standard practice in sensitivity analyses. However, ensuring that models are able to represent physical processes is critical to confidently conclude on the spatial behavior of process sensitivities.

2. *Sensitivity analysis: My main concern is that the selected parameter bounds may be inappropriate (i.e., too wide) for a large fraction of catchments, providing unrealistic sensitivity indices for some processes. Did the authors use the parameters and weights obtained from the calibration process to inform the parameter ranges used in the xSSA method? Or did the authors examine the sensitivity of the results (and conclusions) to the choice of parameter bounds, at least for the catchments in Figure 4? If not, I suggest doing this exercise to ensure that the conclusions presented in this paper are well supported.*

We thank the reviewer for this observation. We recognize that the adequacy of parameter bounds may vary from catchment to catchment. However, it is very important to note that to facilitate analysis of thousands of basins, certain assumptions of uniformity must be made. We here chose to keep the parameter range values fixed across all basins to make the results easier to interpret and the analysis feasible. Importantly, using different ranges for individual basins would also make the definition of sensitivity metrics vary between basins, meaning that we could no longer compare them directly.

To address the reviewers concern regarding sensitivity of results to parameter ranges, we analysed the distributions of the calibrated parameters and identified five that exhibited a non-uniform posteriori distribution (x_2 to x_6); all other calibrated parameters were spread within their prior range specified. We reduced the ranges of those five parameters by 10% on each side (20% range reduction in total) except for parameter x_5 where adjusting the lower bound did not make sense and instead we reduced the upper bound by 20%. We performed the entire analysis for 150 randomly selected basins and compared the process sensitivities without and with range reduction (Fig. R3). As the results show the mean absolute differences for all processes are negligible (at most 0.0129 for ‘Evaporation’). We think this nicely demonstrates that the results are very robust to modest changes in parameter range. We however thank the reviewer and might adjust the ranges for the planned sophisticated calibration experiment informed by this SA study. We included this in the manuscript now:

line 324 ff. To assess the impact of a priori range specification for each parameter (Table S2 in Supplementary Material), we repeated the analysis for a subset of 150 basins using ranges for parameters x_2 to x_6 that were reduced by 20%.

line 93 ff. The sensitivity analysis results were determined to be robust to changes in specified parameter ranges, with negligible (< 0.0135) changes to process sensitivities based an analysis using a subset of 150 randomly selected basins (results not shown).

3. *Statistical model: I think that there are several methodological decisions that need justification and/or further explanation, including:*

Figure R3: The process sensitivities with the original ranges used throughout the presented study (first stack of markers per process) and sensitivities using reduced ranges for 5 parameters (x_2 to x_6 ; second stack of markers per process). The ranges were reduced by 10% on either side of the range (i.e., x_2 , x_3 , x_4 , and x_6) or 20% if only one bound makes sense to adjust (i.e., x_5). Results are shown for 150 randomly chosen basins. The values displayed below the two stacks of markers per process indicated the mean absolute difference between the sensitivities using original and reduced ranges. The gray lines connect the pair of sensitivities per basin.

- *What is the rationale of considering latitude and longitude among the pool of potential predictors? Please provide a reasonable justification, since they do not necessarily explain the spatial variability of catchment responses, which result from the combination of climatic and physiographic characteristics that can be highly heterogeneous in relatively short distances.*

This was a great point. We have hence excluded them now as can be seen in the revised Table 1.

- *Did you start with the simple exercise of plotting your sensitivity indices against your catchment descriptors (i.e., x , xy , y^2 , etc.), and computing correlation coefficients (both Pearson and Spearman) and p -values, in order to justify the statistical model structure? Did you check the multi-collinearity among predictors?*

Yes, that is what we did in the beginning and thought that we could do better than that by performing the multi-variate regression that we applied in the end. This led to significant improvements in the statistical performance metrics (i.e., Pearson correlation). We hope that the approach that we are now using (one predictor and switch to two predictors only if it leads to an improvement of R_{adj}^2 at least 0.05) helps to address this concern. Multi-collinearity was not explicitly checked.

- *Finally, if the goal is to build statistical models for prediction, a cross-validation type metric (like GCV or PRESS) would have been a better choice to select the best model. Additionally, the authors should have used the adjusted R^2 to compare models with different numbers of predictors (instead of the coefficient of determination, R^2).*

We agree. We now perform a cross-validation analysis to select the best model. We did not use the PRESS. We are using the same cross-validation approach as used later in our final calibration/validation experiment: we use two-thirds of the datapoints to do the regression and then cross-validate with the remaining one-third of datapoints. We repeat this 10 times and pick the regression model that has the best average adjusted coefficient of determination. We would like to mention that, the results are very similar to our original approach. We adjusted Table 1, Figure 5 (see Fig. R5 below), as well as the following description in the manuscript:

line 183 ff. The regression model selection was guided by performance in predictive mode assessed in cross-validation experiments.

line 351 ff. The regressions were performed in a cross-validation setup, i.e., each polynomial was fitted to two-thirds of the basins and validated using the remaining one-third of the basins. The polynomial with the best average adjusted coefficient of determination R_{adj}^2 in validation across 10 trials was identified- one best across all one-predictor polynomials and one best across the two-predictor polynomials.

line 359 ff. The regression function leading to the best adjusted coefficient of determination R_{adj}^2 is then used to perform a regression using all 3316 basins (reported in Table I). To assess the robustness of this regression, we performed another cross-validation experiment, similar to the previous one. The regression coefficients of the best regression (reported in Table I) are refit now using two-thirds of the basins and validating them using the remaining one-third of the basins (reported in Fig. 5). This experiment is repeated with 100 random splits of basins.

Besides that, we are using and reporting the adjusted coefficient of determination R_{adj}^2 now to select and evaluate the regression models. It is (basically) the same as R^2 since the number of datapoints used is very large (> 2000 cal, > 1000 val) and the number of predictors is very small (1 or 2). We previously had used the correlation coefficient r . We adjusted the manuscript

Table R1: Statistics regression functions in previous and current version of the manuscript.

Process	Previously				Revised			
	MAE_{cal}	MAE_{val}	ρ_{cal}	ρ_{val}	MAE_{cal}	MAE_{val}	$R_{adj,cal}^2$	$R_{adj,val}^2$
Infiltration	0.015 ± 0.000	0.015 ± 0.000	0.965 ± 0.001	0.964 ± 0.002	0.017 ± 0.000	0.017 ± 0.000	0.906 ± 0.003	0.904 ± 0.005
Quickflow	0.033 ± 0.000	0.033 ± 0.001	0.976 ± 0.001	0.976 ± 0.002	0.033 ± 0.001	0.033 ± 0.001	0.953 ± 0.002	0.952 ± 0.003
Evaporation	0.014 ± 0.000	0.015 ± 0.000	0.981 ± 0.001	0.981 ± 0.001	0.020 ± 0.000	0.020 ± 0.000	0.915 ± 0.003	0.915 ± 0.005
Snow Balance	0.014 ± 0.000	0.014 ± 0.001	0.977 ± 0.001	0.977 ± 0.002	0.019 ± 0.000	0.019 ± 0.000	0.926 ± 0.002	0.926 ± 0.004
Convolution (srfc runoff)	0.001 ± 0.000	0.001 ± 0.000	0.811 ± 0.009	0.808 ± 0.018	0.001 ± 0.000	0.001 ± 0.000	0.657 ± 0.017	0.649 ± 0.036
Potential Melt	0.027 ± 0.000	0.027 ± 0.001	0.984 ± 0.001	0.983 ± 0.001	0.033 ± 0.000	0.033 ± 0.001	0.950 ± 0.002	0.950 ± 0.003
Percolation	0.014 ± 0.000	0.014 ± 0.000	0.965 ± 0.000	0.965 ± 0.002	0.016 ± 0.000	0.016 ± 0.000	0.909 ± 0.003	0.909 ± 0.005
Rain-Snow partitioning	0.026 ± 0.001	0.026 ± 0.001	0.903 ± 0.003	0.903 ± 0.007	0.026 ± 0.001	0.026 ± 0.001	0.815 ± 0.007	0.812 ± 0.014
Precipitation Correction	0.007 ± 0.000	0.007 ± 0.000	0.911 ± 0.003	0.912 ± 0.006	0.008 ± 0.000	0.008 ± 0.000	0.809 ± 0.006	0.805 ± 0.013

Note. The results of the regression analysis in the previous version of the manuscript and the current version. The current version does not use ‘lat’ and ‘lon’ as predictors anymore, reports R_{adj}^2 , uses only one predictor if two predictors does not improve the R_{adj}^2 by more than 0.05, and is using a cross-validation approach to obtain the best regression model. All results are based on 66% of the 3316 datapoints in “calibration” (i.e., performing regression) and 33% of the datapoints in “validation” (i.e., testing regression function).

and report R_{adj}^2 now in Tab. 1 and Fig. 5 and throughout the manuscript.

Since we now a) using 1 predictor by default and only use 2 predictors if the R_{adj}^2 improves by more than 0.05, b) excluding ‘lat’ and ‘lon’ as predictors, c) reporting R_{adj}^2 instead of r and d) using the cross-validation approach as described above, the nominal results changed slightly. The conclusions however are the same as in our previous manuscript. A summary of the statistics in the previous version of the manuscript vs. the results reported now in Fig. 5 are shown in the table below. The updated Fig. 5 shown now in the manuscript is displayed for reference below as Fig. R5.

Specific comments

4. L8-9: I would be careful with this statement, since the sensitivity of any hydrological variable can be quantified, more generally, by perturbing factors, which may also include input forcing variables such as precipitation or temperature (e.g., Raleigh et al. [2015]).

We absolutely agree with this. In our definition these ‘perturbing factors’ are also parameters- in the sense that they are ‘tunable items’ in a model calibration. We do not want to distinguish between ‘factors’, ‘coefficients’, ‘variables’, and ‘parameters’ (see also reply to comment #22). We indeed analyse such ‘perturbing factors’ in our analysis as well: rain corrector factor (x_{33}) and snow correction factor (x_{34}). We clarify what we define as ‘parameters’ now early in the manuscript:

line 31 ff. Note that parameters can be (traditional) model parameters, multiplicative factors to perturb input forcings, or parameters to weight between different options, among others.

5. L16: From my understanding of the paper, the estimation of ‘entire process sensitivities’ includes parameters (right?). If yes, I think it would be good to clarify here.

Yes, the reviewer is right and we wish we could address this but we reached the word limit of the abstract and were unable to effectively revise it to include this detail. We however hope that it becomes more clear throughout the manuscript.

Figure R4: **Attempt to revise Figure 5.** Same as original Figure 5 but using only basins with NSE greater than 0.7 rather than basins with NSE greater than 0.5 as used originally. The functional relationships and all statistical measures reported are unchanged.

Figure R5: **Revised Figure 5.** With a) not using latitude and longitude as predictors anymore, b) reporting adjusted coefficient of determination R^2_{adj} instead of correlation coefficient r (previously ρ), c) using one predictor if two predictors do not improve R^2_{adj} by at least 0.05 and d) using a cross-validation approach to determine the regression function used.

6. *L21: I don't think that reference [2] is appropriate in the context of hydrological models for flood forecasting (indeed, Cao and Zhang [2016] do not mention hydrological models at all), and the authors could cite more relevant papers in this field (e.g., Weerts and El Serafy [2006]; Todini [2008]; Bogner and Pappenberger [2011]; Rakovec et al. [2015]).*

Thanks a lot for pointing this out and providing those more relevant publications. We adjusted the manuscript accordingly:

line 21 ff. Hydrologic models are widely used in applications that are important for society such as flood prediction [Weerts and El Serafy, 2006, Todini, 2008, Bogner and Pappenberger, 2011, Yucel et al., 2015, Rakovec et al., 2015, Rogelis and Werner, 2018], [...]

7. *L23-24: I agree, but the authors should also recognize that a thorough model evaluation that takes advantage of information theory and newly available datasets is critical to advance hydrologic process understanding, and this is a good place to mention this, with proper citations (e.g., Gupta et al. [2008]; Hrachowitz et al. [2014]; Nijzink et al. [2018]; Khatami et al. [2019]; Széles et al. [2020]; Dembélé et al. [2020]).*

We agree and adjusted the manuscript as follows including the suggested citations.

line 24 ff. Further developments and improvements of hydrologic models are essential to advance the understanding of hydrologic processes and ensure greater model realism [Coron et al., 2014, Clark et al., 2017, Menard et al., 2020, Mai et al., 2021]. One way such improvements can be ensured is by carrying out model evaluations taking advantage of information theory and newly available datasets [Gupta et al., 2008, Hrachowitz et al., 2014, Nijzink et al., 2018, Khatami et al., 2019, Széles et al., 2020, Dembélé et al., 2020]. Sensitivity analysis (SA) is a well-established tool to guide such model assessments [Mendoza et al., 2015], [...]

8. *L26-28: in a broader sense, SA is used to identify the most sensitive factors, which could be more than just model parameters (see comment #4). It is also confusing that the authors refer to 'parametric' models, since the paper is about process-based hydrological modeling (which contain many parameters).*

We apologize for this oversight. We were looking for a phrase to name 'models that contain parameters that need to be inferred by, for example, model calibration'. 'Parametric model' (aka a model that contains parameters) was not the right choice. We adjusted the manuscript accordingly:

line 29 ff. SA is [...] a general method that can be applied to any kind of model that contains unknown parameter estimates [Saltelli et al., 2008, Ferretti et al., 2016, Saltelli et al., 2019]. Note that parameters can be (traditional) model parameters, multiplicative factors to perturb input forcings, or parameters to weight between different options, among others.

9. *L28: I'm not sure that Konapala et al. [2020] should be listed here, since they did not conduct formal sensitivity analyses.*

We agree. We removed that citation.

line 27 ff. Sensitivity analysis (SA) is a well-established tool to [...] identify the most critical relationships within a system [Göhler et al., 2013, Cuntz et al., 2016, Markstrom et al., 2016].

10. *L29: 'SA is a well-established method'. Perhaps it would be more appropriate using the word 'tool', since there are many SA methods.*

We adjusted this.

line 33 ff. Notwithstanding the repute of SA as a tool, there are several challenges [...]

11. *Figure 1: it is common practice to start hydrological analyses (especially with numerical, process-based models) at the beginning of the water year, when catchments reach the ‘minimum’ water storage. Please provide an explanation of why you decided to start on January 1st.*

We understand that this might be common practice especially for calibration exercises. However, we decided to start at the beginning of the calendar year as this is less important for sensitivity analyses that focus on changes in the model outputs and its variability rather than nominal values itself. We also believe the 2-year spin-up period addresses any potential impacts of using a calendar year rather than water year start date by allowing storage variables to reset prior to the analysis period.

12. *L79-81: does ‘in agreement with expectations’ only refer to snow balance, or in general to all processes?*

This holds for all processes. It is the first very general observation that can be made when looking at the sensitivity results. We make this statement as a first qualitative evaluation of the results. We added another example to confirm that we mean each process; not only snow balance. We also replaced ‘expectations’ with ‘hydrologic reasoning’ that we do not mean (quantitative) ‘expectations’.

line 91 ff. The patterns are in agreement with hydrologic reasoning; for example, snow balance sensitivity is high in mountainous and northern regions and potential melt is only sensitive where snow occurs.

13. *L102: ‘The analyses presented here [...] provide insights over several model structures’. I would delete statements like this from the manuscript, since the results do not support clear recommendations on which model structures (i.e., dominant processes, process parameterizations, model architecture, spatial connectivity, etc.) are more appropriate in different regions.*

We think it is important to mention here that instead of analysing a fixed model structure as Markstrom et al. [2016], we performed an analysis over a wide range of possible model structures (108 base models and all continuous combinations of those models; see Supplementary Material “Details about the Blended Model”), as this is a unique contribution of this work. We added the term ‘averaged’ to make more clear what we mean here.

line 117 ff. The analysis herein furthermore represents an improvement upon the Markstrom et al. [2016] study, as it provides insights averaged over several model structures, [...]

14. *L114: please provide a reference for c-means fuzzy clustering.*

Done.

line 130 ff. The regions are obtained by a c-means fuzzy clustering [Bezdek, 1981] based on [...]

15. *L148: the dark green for baseflow is really hard to see.*

We apologize for this. It is however difficult to find 11 colors that everyone can clearly distinguish. We feel in case of baseflow this is negligible since it is a process of such a weak overall sensitivity.

16. *L148-149: I do not see quick flow as ‘almost constant’ in Figure 4f. Did you look at other basins in the same cluster?*

We adjusted this by adding an additional note about quickflow in the manuscript:

line 164 ff. The importance of infiltration (light blue) and quickflow (medium blue), however, is almost constant throughout the entire year. The latter shows a slightly decreased sensitivity during the melt period (March and April) when other processes become more important.

Basins within this cluster behave reasonably similar as can be checked on our website. We however wish to emphasize that we derived the clusters and chose the representative basins solely with the background to have a proper reason for which example basins to show. We assumed that if we would have just picked basins randomly, this selection would have been questioned.

17. *L166: I suggest changing the Greek letter currently being used by 'r', since rho is normally reserved for the Spearman rank correlation coefficient.*

We thought it would be easier to identify the symbol ρ within the manuscript text than r but we absolutely agree with the reviewer. We however do not use the correlation coefficient r anymore but the coefficient of determination R_{adj}^2 following another suggestion of the reviewer. Therefore this change is obsolete. Please see, for example, **Figure 5** and **Table 1**) for the proper use of the new symbols.

18. *L166: I don't think that 'reliability' is the right word for this context, since it typically refers to an attribute of probabilistic simulations or predictions. I would recommend using 'stability' or a similar word.*

We agree and adjusted the manuscript as follows hoping that it is less confusing now:

line 185 ff. To obtain a measure of how sensitive these functions are to the choice of training basins, the basin set is split into 100 random subsets of basins, [...]

19. *L168: Please clarify if you used all 3,316 catchments to derive the coefficients in Table 1. What period did you use to compute climate attributes?*

We agree that these are important details that should be mentioned here. We resolved this as suggested.

Note below Table I Note. Results of the regression analysis identifying the functional relationships between predictors and the variance-weighted total Sobol' sensitivities ST_i^w based on the results of the 3316 basins analyzed by xSSA. The regression is performed using polynomials with one predictor (up to degree three with at most six coefficients) and polynomials with two predictors (up to degree three with at most five coefficients). The one-predictor polynomial is used unless the use of a pair of predictors led to an improvement of the adjusted coefficients of determination R_{adj}^2 by at least 0.05. The basin characteristics used as predictors are derived from geophysical attributes and the meteorology available for each basin (1950 to 2010).

20. *L197-198: It is really hard to believe that altitude does not play a considerable role here, but longitude does (see major comment #3).*

We were surprised by this as well. The issue is however resolved now as we revised the approach to obtain the functional relationships as the reviewer had suggested. The latitude and longitude are not used as predictors anymore and we only use one predictor unless the inclusion of a second predictor leads to an improvement of the adjusted coefficient of determination R_{adj}^2 of at least 0.05. See also reply to the major comment #3.

21. *L225: do you actually need thousands of basins for deriving these ‘functional relationships’?*

We agree with the reviewer that thousands of basins might not be necessary to derive the relationships. One might be able to infer robust relationships with a smaller set of basins as long as this set of basins is representative of the climatic, physiographic, and other drivers that determine the process sensitivities. We did not look into reducing the set of basins besides the validation experiments where we only used two-thirds of the basins to infer the relationship and test them at the remaining one-third of basins. It would be interesting if the same relationships would be inferred if only, for example, the CAMELS basins would have been used. We however decided to go the safest route and use all available information to derive those relationships. We mention in the ‘Discussions’ that results could be based on smaller sets of basins (text snippet was already included in the previous version of our manuscript):

line 267 ff. Our approach to continental-scale sensitivity analysis can be emulated for more complex land-surface hydrologic models, and the results suggest such an analysis could be conducted over a reduced number of watersheds.

22. *L231: ‘reducing the risk of inferring model-specific conclusions’. I disagree with this statement, since such ‘risk’ should be minor if you select your model based on careful model structure diagnostics using field observations and in situ measurements. Additionally, the concept of ‘blended model’ does not really help to reduce structural and parametric uncertainties if model evaluation is simply conducted by NSE or KGE. I think the authors should recognize this limitation, and also acknowledge a large body of work based on inference from observations to build hypotheses in hydrological modeling (e.g., McMillan et al. [2011]; McMillan et al. [2014]; Clark et al. [2011]; Fenicia et al. [2014]; Fenicia et al. [2016]) and work towards ‘the right answers for the right reasons’ (Kirchner [2006]).*

We think there is several points here:

- (a) We respectfully disagree with the reviewer in our understanding of ‘reducing the risk of making model specific conclusions’. Objectively, this risk is reduced as we analyse a range of models (108 base models and all continuous combinations of those models; see Supplementary Material “Details about the Blended Model”) rather than presenting the results of one specific model. Therefore, there is less of a risk of finding model-dependent results.
- (b) Following this, it is trivial that there would be no model structural uncertainty if one would know the correct model structure to implement a priori. However, one of the aims of SA is to help find which processes to implement and focus on in the model selection and development, therefore it is added information to improve that choice. the same argument can be made for other decisions that need to be made to build a model: meteorological inputs and parameter selections would also lead to better modelling systems if we knew a priori which datasets and parameters to use.
- (c) The xSSA analysis investigates if a choice of a process option matters or not (process option sensitivities) and on which process a modeller should focus when simulating streamflow (process sensitivities).
- (d) We wish to emphasize that the xSSA analysis is neither based on NSE nor KGE (or any metric of model performance). It analyses the simulated streamflow time series itself. Hence the sensitivities regarding alternative hypotheses could easily be derived from the data we provide through this project.

We however included the suggested citations now to acknowledge this relevant body of work.

line 246 ff. The use of the Raven hydrologic modeling framework [Craig et al., 2020] and the novel blended model structure of [Mai et al., 2020] allow a seamless analysis of a range of model structures, thereby reducing the risk of inferring model-specific conclusions as a range of model structures are analyzed simultaneously. It should be noted that the model structure can be inferred during model calibration based on observations, to build hydrological modeling hypotheses (e.g., McMillan et al. [2011], McMillan et al. [2014], Clark et al. [2011], Fenicia et al. [2014], and Fenicia et al. [2016]). This work can help modelers decide on the model structure to use for inference of model structure in the following manner. First, processes with small sensitivities can be represented with essentially any option since the streamflow simulations are not affected in an important manner. Second, the processes with larger sensitivities should be included in the calibration process since the selection of the process option will play an important role in the streamflow simulations.

23. *L234: although I commend the authors for the fantastic outcome (a very nice website with a lot of information), I think it would tremendously benefit from including more graphics (time series with simulated and observed streamflow, scatterplots, simulated and observed flow duration curves, seasonal runoff, etc.) of calibration and validation model results.*

We agree with the reviewer that the website would benefit from additional graphics etc. We however think we already went significantly beyond what would be considered the state-of-the-art for reporting analyses like this by creating a website with the current functionality. All the additional functions the reviewer mentions focus on the calibration results and not the sensitivity analysis, which is the focus of this publication. We have decided to visualize only results directly related to the work presented here.

We however would like to note that all data necessary to produce the requested hydrographs etc. are shared on the website and can be downloaded and produced if interested.

24. *L256: ‘eight parameters’. I assume that you are referring to r_1 , r_2 , etc., which define the weights in Raven. Perhaps it would be better referring to them as ‘coefficients’ instead.*

We think the distinction of ‘parameters’ and ‘coefficients’ might be more confusing than helpful. The parameters r_1 , r_2 , etc are indeed used to derive the weights (in a non-trivial way) but their handling within the sensitivity analysis is in no way different than for any other parameter/coefficient. Besides this, the parameters x_{33} and x_{34} are rain and snow correction factors, respectively. If we would distinguish between ‘coefficients’ and ‘parameters’, we would need to apply this terminology also to parameters like this and call them ‘coefficients’. Again, we think this would be more confusing than helpful. We now give examples for our definition of parameters in the manuscript:

line 31 ff. Note that parameters can be (traditional) model parameters, multiplicative factors to perturb input forcings, or parameters to weight between different options, among others.

25. *L259: I think that the manuscript should be self-contained and, at the very least, a better description and list of parameters (including and justifying the choice of ranges) should be included.*

We agree and added the section “Details about the Blended Model” to the

Supplementary Material. Most of the content (parameter tables, flowcharts, etc) are the same as in the publication introducing the Blended Model [Mai et al., 2020]. We mention those additional resources now in the manuscript:

line 282 ff. The specific process options and associated model parameters used here are available in Tables C1 and C2 of Mai et al. [2020], and are added to the Supplementary Material of this publication for the convenience of the reader.

line 289 ff. More details about the model, its parameters and processes can be found in the Supplementary Material as well as in Mai et al. [2020].

Did you implement the models in lumped mode, as Mai et al. [2020] and Chlumsky et al. [2021]?

We apologize that this was not clear enough. Yes, we used exactly the same setup etc as in the two previous publications. We added the following to the manuscript to emphasize this stronger.

line 288 ff. The model setup used in this study is exactly the same as in Mai et al. [2020], except that here, it is applied to more than 3000 basins.

Did you calibrate 35 + 8 parameters in all catchments?

Yes, we did. We note this now more explicitly in the manuscript:

line 305 ff. A budget of 2000 model evaluations per basin and ten independent trials is used for DDS to calibrate the 35 model parameters and the 8 parameters defining the model structure.

We also added a note to the potential question of overfitting this large number of parameters might raise:

line 312 ff. We assume that a strong validation performance indicates a low likelihood that the model parameters are overfitted during calibration.

26. *L271-272: Did you find large differences between parameter sets and NSEs from the 10 trials, to justify doing this for more than 3,000 catchments?*

We solely applied a ‘calibration’ to be more efficient than a pure random search. The algorithm used here is DDS which is known to find good solutions with a small budget. In our experience it has more advantage to deploy 10 trials with budget X than one run with 10X budget. A budget of 2000 is likely not enough for ‘converged’ calibration results and hence we expect a spread of the 10 different results. We did not analyse this as we are only interested in the potential performance of the applied model. We think we demonstrate adequate performance across the set of watersheds.

27. *L280: ‘demonstrated at one example catchment per climate zone’. This is confusing. I understand that you applied the method for all catchments, and NOT a few example basins. Perhaps a better word would be “illustrated”.*

Done.

line 319 ff. [...] illustrated at one example catchment per climate zone.

28. *References: I’m not sure that the format is correct, since it is not possible to see the title of the papers.*

We apologize that the references were formatted like this. We changed the bibliography style and the paper titles should appear now.

Suggested edits

29. L11: *'based on basins'* reads awkward. Perhaps it would be better using *'based on catchments'*.
We rewrote the entire abstract. This sentence does not exist anymore.

30. L11: *'enabling the estimation of the process sensitivities'* → *'to estimate process sensitivities'*.
Done.

line 12 ff. This enables seamless quantification of model process sensitivities and parameter sensitivities across a continuous set of models.

31. L12: *'needing'* → *'the need'*.
Done.

line 17 ff. [...] without the need to perform expensive sensitivity analyses.

32. L13: delete *'itself'*.
Done.

line 17 ff. [...] without the need to perform expensive sensitivity analyses.

33. L23: delete *continuously*.
Done.

line 23 ff. [...] the underlying complexity of hydrologic processes is leading to increasing complexity among these models [...]

34. L32 and everywhere else: I suggest using *'indices'* instead of *'indexes'*.
Done at multiple location, for example:

line 36 ff. [...] sensitivity indices of parameters on components or processes of the water cycle [...]

35. L32: *'hydrologic cycle'* → *'water cycle'*.
Done.

line 36 ff. [...] sensitivity indices of parameters on components or processes of the water cycle [...]

36. L47: *'structure'* → *'structures'*.
Done.

line 51 ff. [...] which enables a seamless analysis of model parameters and model structures, [...]

37. Caption of Figure 1: *'The reasons for reducing the sets of basins from the original 14 425 basins in the HYSETS database to 3316 basins'*. I would delete the third *'basins'* (and perhaps also the second).
Done.

New caption Figure 1 The reasons for reducing the number of basins from the original 14,425 in the HYSETS database to 3316 for the xSSA sensitivity analysis [...]

38. L67: delete *'here-used'*.
Done.

line 72 ff. The weaker performance of the blended model during validation over the high plains [...]

39. L69: *‘are not considered’* → *‘are not included’*.

Done.

line 74 ff. [...] and the basins are not included in the analyses to follow.

40. L128-150: *I recommend writing the full names for all months.*

Done at multiple places between **lines 141 and 173**.

41. L174: *‘the ones that are’* → *‘those’*.

Done.

line 193 ff. All panels show that the predicted sensitivities are in close agreement with those derived by the xSSA analysis, [...]

42. L180: *insert comma after ‘this process’*.

Done.

line 198 ff. Only 88 of the 3316 basins show sensitivities larger than 0.01 for this process, which could explain the reduced overall predictability.

43. L185-186: *I suggest placing the following text between hyphens or commas: ‘already identified to be important in the Rocky Mountains and the higher latitudes starting over the Great Lakes (see Fig. 3)’*.

Done.

line 203 ff. The “Potential melt” (Fig. 5f), “Precipitation correction” (Fig. 5i), and “Snow balance” (Fig. 5d) processes, already identified to be important in the Rocky Mountains and the higher latitudes starting over the Great Lakes (see Fig. 3), are also reliably quantified, even for the rarer large sensitivity indices.

44. L200: *‘wetness and presence of snow are both key indicators’* → *‘both wetness and presence of snow are key indicators’*.

This does not exist in the revised manuscript anymore since we changed the regression models addressing another suggestion of the reviewer.

45. L201: *‘a best predictor’* → *‘the best predictor’*.

The manuscript changed due to the changes in the regression models, which makes the suggested revision obsolete.

46. L202: *insert comma after ‘in extreme cases’*.

Done.

line 219 ff. In extreme cases, the regressions can result [...]

47. L208: *‘indicates robustness of the results and the transferability’* → *‘indicates their robustness’*. I suggest deleting *‘transferability’*.

Done.

line 225 ff. The consistency of results with previous studies indicates their robustness with respect to other models (e.g., PRMS) and sensitivity metrics (e.g., FAST method based on mean streamflow).

48. L211: ‘*This large-scale continental sensitivity analysis*’ → ‘*This continental-scale sensitivity analysis*’.

Done.

line 228 ff. This continental-scale sensitivity analysis of hydrologic model parameters and processes addresses [...]

49. L222: *insert comma after ‘Mountains’.*

Done.

line 239 ff. In snow-dominated regions such as the Rocky Mountains, the “Potential melt”, “Snow balance”, “Precipitation correction”, and “Rain-snow partitioning” processes are of large importance.

50. L225: ‘*analyses*’ → ‘*analyzed*’.

Done.

line 242 ff. Furthermore, the large number of basins analyzed allows to deduce functional relationships [...]

51. L242-243: ‘*approximation of the sensitivity of hydrologic processes on streamflow simulations*’. I suggest re-writing as ‘*sensitivity of streamflow simulations to hydrologic processes*’.

Done.

line 271 ff. This study is the first quantitative continental sensitivity analysis of streamflow simulations to hydrologic processes [...]

Reviewer #3:

In this study, the authors conducted a very comprehensive hydrologic model sensitivity analysis using the Raven hydrologic modeling framework for over 3000 basins across the North America. The authors rigorously calibrated, validated, and selected qualified basins to support their analysis, and then concisely reported their sensitivity analysis findings by major hydrologic processes in space and time. The manuscript was nicely written, clearly organized, and provided useful scientific insights to the broader hydrologic modeling community for the development and enhancement of future hydrologic models. Based on what I know, this is by far the most elaborated and comprehensive hydrologic model sensitivity analysis in the North America, and hence provides timely importance for a highly impactful journal like Nature Communications. I would gladly recommend accepting this manuscript, after the following concerns have been satisfactorily addressed in the revised manuscript.

We thank the reviewer for the positive evaluation of our manuscript and the kind words. We provide a detailed reply to all comments raised below and hope that, especially, the concerns regarding the “calibration” are more clear now.

1. *While I agree with the flexibility of Raven, it is still a form of model-specific, multi-structural hydrologic models (like FUSE and SUMMA). It does provide alternative hydrologic processes for different location-specific needs, but is also unique to its non-structural features such as software workflow, numerical solvers, and spatial-temporal meshing strategies. In addition, unless you considered the uncertainties associated with non-calibrated land surface parameters and meteorologic forcing datasets, your findings should also be specific to your applications. In my opinion, the authors should provide additional caveats and discuss the potentially different findings when using other multi-structural hydrologic models, land surface parameter sets, meteorologic observations, and hydrologic modeling strategies.*

We agree that there is likely some influence of the choice of multi-model (Raven), observation datasets, and the specific blended model configuration we applied here (which determines the relevant parameters). However, it is very difficult to speculate (without an immense amount of additional work) what these impacts may be, and feel that it is not appropriate to do so without hard data. We are more comfortable in discussing the potential challenges in deploying the approach for other models, observation sets, and modelling strategies. We added the following to the discussion:

line 265 ff. It would additionally be interesting to investigate the influence on the sensitivities when other forcing datasets are used or other modeling decisions - such as different numbers of soil layers, other process options, or different simulation time steps - are made.

The reviewer further mentions additional ‘non-structural features’ that are unique to Raven. We think that these exist in any model. To our understanding the ‘model workflow’ should not impact the result of a hydrologic simulation. Raven employs a stable ‘numerical solver’ as shown in Snowdon [2010] and in Craig et al. [2020]. ‘Spatial’ meshing will likely always be a model specific setting while ‘temporal’ meshing (i.e., simulation timestep) could be easily added as a parameter in the sensitivity analysis.

We agree that the potential of performing an xSSA analysis with other modeling frameworks should be mentioned in the manuscript. We added the following:

line 335 ff. The xSSA analysis is performed using the blended model setup within the Raven modeling framework [Craig et al., 2020], but can also be performed with any modeling framework, such as SUMMA [Clark et al., 2015a,b] and

FUSE [Clark et al., 2008]. The process sensitivities can be estimated as soon as parameters are grouped into processes (no model implementation required), while properly analyzing multiple process options simultaneously requires the implementation of the weighted averages of these process option outputs (implementation in model most likely required).

2. *There seems to be a large number of model parameters in Raven (35 model parameters and 8 process combinations). How can one ensure that we have sufficiently long historical observations for calibration? Have you also evaluated other non-streamflow observations such as snow or evapotranspiration? In other words, how can one ensure that this is not an underdetermined question? Please comment on this challenge in the revised manuscript.*

We think the evaluation of undertermination is important for any calibration study, but we once again clarify that this paper describes calibration experiments primarily to exclude incapable models and to demonstrate basic adequacy of these models (as noted in responses to reviewer #2). We assess the determination question via validation, assuming that strong validation performance indicates a low probability of overfitting; by definition, overfit models are poor extrapolators. We have made the plan of future continental scale calibration experiments (where we do intend to address some of these issues in a more rigorous manner) more clear.

We added the following to emphasize the aim of the preliminary calibration/validation:

line 299 ff. A preliminary calibration/validation experiment of the blended model is performed to demonstrate the basic adequacy of the recently introduced blended model, and to act as the basis for excluding poorly performing models from analysis. This basic calibration using a single evaluation metric will be the precursor to a more elaborate multi-metric model calibration/validation study to be informed by the results found here.

We also added text to highlight the issue of undertermination:

line 312 ff. We assume that a strong validation performance indicates a low likelihood that the model parameters are overfitted during calibration.

3. *It is interesting to see that the authors found unclear sensitivities associated with the convolution processes which control the timing of runoff. I am wondering to what level this may be affected by the timing issues within the meteorologic observations? To be more specific, a recent study by Pierce et al. (2021) suggested that the timing adjustment used in the Livneh et al. (2015) dataset (used in HYSETS) may be unsuitable and can lead to underestimation of hydrologic extremes. This issue was caused by the varying reporting time of daily rainfall stations, which usually reported daily rainfall from 7am to 7am instead of midnight to midnight. In other words, how was the rainfall timing issue addressed in HYSETS? Do the meteorologic observations in HYSETS have sufficiently accurate timing to help you evaluate the sensitivity of hydrologic processes associated with runoff timing?*

We agree with the reviewer that shifts in meteorologic forcings impact model performance- especially when they lead to shifts in extreme events. This however is with respect to model performance; our sensitivity estimates are not calculated with respect to performance metrics but directly on the simulated streamflow, which is independent of the observation data. If there is a shift error in the meteorologic forcings this would not impact the results in any way since all the simulated time series would see the same shift. However, because all of the sensitivity metrics are based upon the magnitude of streamflow (rather than its timing), the most sensitive processes will be the ones which most greatly impact the streamflow magnitude, not the timing. We have clarified this in the text:

line 106 ff. It is not surprising that these three processes (convolution of surface and delayed runoff and baseflow) are less sensitive since the sensitivity analysis assesses the variability in streamflow magnitudes rather than its timing.

We are not too surprised to see only low sensitivities of the routing components as we use lumped model setups here. The sensitivities are likely to increase when distributed model setups are used.

We thank the reviewer though for pointing us to this issue with the HYSETS dataset. We will likely use another forcing [Gasset et al., 2021] when targeting the proper continental calibration of the blended model.

4. *The study built on Chlumsky et al. (2021), which reported detailed calibration findings of 12 MOPEX12 catchments. Since the number of basins has been largely expanded in this study (to over 3000), I think the readers would also like to see some further information from this more comprehensive calibration. For instance, among the alternative hydrologic processes, which ones typically have the largest weights across over 3000 basins?*

This study is actually building on the work presented in Mai et al. [2020] (i.e., sensitivity analysis of model structure) and extending that work to basins across North America. We intend to also extend the work of Chlumsky et al. [2021] to this set of basins and analyse exactly what the reviewer is suggesting. The work presented here will inform this elaborate calibration experiment. The ‘calibration’ performed here is solely to confirm that the fairly new blended model has skill across the continental domain studied here. We thought this might be informative as this model had otherwise never been applied over this domain.

We apologize for the confusion this might have initially caused and use the terms ‘preliminary calibration/validation’ now throughout the manuscript. A few examples are the following:

line 62 ff. Preliminary Calibration and Validation

line 63 ff. A preliminary calibration/validation experiment of the blended model is performed to demonstrate the basic adequacy of the recently introduced blended model, and to act as the basis for excluding poorly performing models from analysis. This basic calibration using a single evaluation metric will be the precursor to a more elaborate multi-metric model calibration/validation study to be informed by the results found here.

line 78 ff. It is important to note that the results of this calibration exercise are used to (1) exclude clearly low quality models from further analysis and (2) to demonstrate the basic adequacy of the models for simulating streamflow over the range of simulated conditions. While a more elaborate calibration study may improve individual optimal model performances, it is unlikely to yield improved global sensitivity estimates. The exclusion of low quality models is admittedly not standard practice in sensitivity analyses. However, ensuring that models are able to represent physical processes is critical to confidently conclude on the spatial behavior of process sensitivities. Detailed results for all basins calibrated and validated, including the calibrated model setups, can be found on the website [Juliane Mai, 2021] associated with this publication.

References

- N. Addor, G. Nearing, C. Prieto, A. J. Newman, N. Le Vine, and M. P. Clark. A Ranking of Hydrological Signatures Based on Their Predictability in Space. Water Resources Research, 54: 1–21, 2018.
- J. C. Bezdek. Pattern recognition with fuzzy objective function algorithms. Advanced applications in pattern recognition. Plenum Press, New York, 1981. ISBN 0306406713.
- K. Bogner and F. Pappenberger. Multiscale error analysis, correction, and predictive uncertainty estimation in a flood forecasting system. Water Resources Research, 47:W07524, June 2011.
- Z. Cao and D.-L. Zhang. Analysis of missed summer severe rainfall forecasts. Weather and Forecasting, 31(2):433–450, 2016. doi: 10.1175/WAF-D-15-0119.1.
- R. Chlumsky, J. Mai, J. R. Craig, and B. A. Tolson. Simultaneous Calibration of Hydrologic Model Structure and Parameters Using a Blended Model. Water Resources Research, 57(5): e2020WR029229, 2021.
- M. P. Clark, A. G. Slater, D. E. Rupp, R. A. Woods, J. A. Vrugt, H. V. Gupta, T. Wagener, and L. E. Hay. Framework for Understanding Structural Errors (FUSE): A modular framework to diagnose differences between hydrological models. Water Resources Research, 44(12):2135, Aug. 2008.
- M. P. Clark, H. K. McMillan, D. B. G. Collins, D. Kavetski, and R. A. Woods. Hydrological field data from a modeller’s perspective: Part 2: process-based evaluation of model hypotheses. Hydrological Processes, 25:523–543, Jan. 2011.
- M. P. Clark, B. Nijssen, J. D. Lundquist, D. Kavetski, D. E. Rupp, R. A. Woods, J. E. Freer, E. D. Gutmann, A. W. Wood, L. D. Brekke, J. R. Arnold, D. J. Gochis, and R. M. Rasmussen. A unified approach for process-based hydrologic modeling: 1. Modeling concept. Water Resources Research, 51(4):2498–2514, Apr. 2015a.
- M. P. Clark, B. Nijssen, J. D. Lundquist, D. Kavetski, D. E. Rupp, R. A. Woods, J. E. Freer, E. D. Gutmann, A. W. Wood, D. J. Gochis, R. M. Rasmussen, D. G. Tarboton, V. Mahat, G. N. Flerchinger, and D. G. Marks. A unified approach for process-based hydrologic modeling: 2. Model implementation and case studies. Water Resources Research, 51(4):2515–2542, Apr. 2015b.
- M. P. Clark, M. F. P. Bierkens, L. Samaniego, R. A. Woods, R. Uijlenhoet, K. E. Bennett, V. R. N. Pauwels, X. Cai, A. W. Wood, and C. D. Peters-Lidard. The evolution of process-based hydrologic models: historical challenges and the collective quest for physical realism. Hydrology and Earth System Sciences, 21:3427–3440, 2017.
- L. Coron, V. Andréassian, C. Perrin, M. Bourqui, and F. Hendrickx. On the lack of robustness of hydrologic models regarding water balance simulation: a diagnostic approach applied to three models of increasing complexity on 20 mountainous catchments. Hydrology and Earth System Sciences, 18:727–746, 2014.
- J. R. Craig, G. Brown, R. Chlumsky, R. W. Jenkinson, G. Jost, K. Lee, J. Mai, M. Serrer, N. Sgro, M. Shafii, A. P. Snowdon, and B. A. Tolson. Flexible watershed simulation with the Raven hydrological modelling framework. Environmental Modelling & Software, 129:104728, July 2020.
- M. Cuntz, J. Mai, L. Samaniego, M. P. Clark, V. Wulfmeyer, O. Branch, S. Attinger, and S. Thober. The impact of standard and hard-coded parameters on the hydrologic fluxes in the Noah-MP land surface model. Journal of Geophysical Research: Atmospheres, pages 1–25, 2016.

- M. Dembélé, N. Ceperley, S. J. Zwart, E. Salvatore, G. Mariethoz, and B. Schaefli. Potential of satellite and reanalysis evaporation datasets for hydrological modelling under various model calibration strategies. Advances in Water Resources, 143:103667, Sept. 2020.
- F. Fenicia, D. Kavetski, H. H. G. Savenije, M. Clark, G. Schoups, L. Pfister, and J. Freer. Catchment properties, function, and conceptual model representation: is there a correspondence? Hydrological Processes, 28:2451–2467, 2014.
- F. Fenicia, D. Kavetski, H. H. G. Savenije, and L. Pfister. From spatially variable streamflow to distributed hydrological models: Analysis of key modeling decisions. Water Resources Research, 52(2):954–989, Feb. 2016.
- F. Ferretti, A. Saltelli, and S. Tarantola. Trends in sensitivity analysis practice in the last decade. Science of the Total Environment, The, 568:666–670, 2016.
- N. Gasset, V. Fortin, M. Dimitrijevic, M. Carrera, B. Bilodeau, R. Muncaster, É. Gaborit, G. Roy, N. Pentcheva, M. Bulat, X. Wang, R. Pavolic, F. Lespinas, and D. Khedhaouria. A 10 km North American Precipitation and Land Surface Reanalysis Based on the GEM Atmospheric Model. Hydrology and Earth System Sciences Discussions, pages 1–50, 2021.
- M. Göhler, J. Mai, and M. Cuntz. Use of eigendecomposition in a parameter sensitivity analysis of the Community Land Model. Journal of Geophysical Research: Biogeosciences, 118(2):904–921, June 2013.
- H. V. Gupta, T. Wagener, and Y. Liu. Reconciling theory with observations: elements of a diagnostic approach to model evaluation. Hydrological Processes, 22:3802–3813, Aug. 2008.
- H. V. Gupta, H. Kling, K. K. Yilmaz, and G. F. Martinez. Decomposition of the mean squared error and NSE performance criteria: Implications for improving hydrological modelling. Journal of Hydrology, 377(1-2):80–91, 2009.
- M. Hrachowitz, O. Fovet, L. Ruiz, T. Euser, S. Gharari, R. Nijzink, J. Freer, H. H. G. Savenije, and C. Gascuel-Oudou. Process consistency in models: The importance of system signatures, expert knowledge, and process complexity. Water Resources Research, 50:7445–7469, Oct. 2014.
- Juliane Mai. xSSA for North America: Calibration results. http://www.civil.uwaterloo.ca/xSSA/webpage/maps_calibration_map.html, 2021. Accessed: May 3, 2021.
- S. Khatami, M. C. Peel, T. J. Peterson, and A. W. Western. Equifinality and Flux Mapping: A New Approach to Model Evaluation and Process Representation Under Uncertainty. Water Resources Research, 55:8922–8941, Dec. 2019.
- J. W. Kirchner. Getting the right answers for the right reasons: Linking measurements, analyses, and models to advance the science of hydrology. Water Resources Research, 42(3):n/a–n/a, Mar. 2006.
- G. Konapala, S.-C. Kao, and N. Addor. Exploring Hydrologic Model Process Connectivity at the Continental Scale Through an Information Theory Approach. Water Resources Research, 56(10):1–23, 2020.
- J. Mai, J. R. Craig, and B. A. Tolson. Simultaneously determining global sensitivities of model parameters and model structure. Hydrology and Earth System Sciences, 24(12):5835–5858, 2020.
- J. Mai, B. A. Tolson, H. Shen, É. Gaborit, V. Fortin, N. Gasset, H. Awoye, T. A. Stadnyk, L. M. Fry, E. A. Bradley, F. Seglenieks, A. G. Temgoua, D. G. Princz, S. Gharari, A. Haghnegahdar, M. E. Elshamy, S. Razavi, M. Gauch, J. Lin, X. Ni, Y. Yuan, M. McLeod, N. B. Basu, R. Kumar, O. Rakovec, L. Samaniego, S. Attinger, N. K. Shrestha, P. Daggupati, T. Roy, S. Wi, T. Hunter, J. R. Craig, and A. Pietroniro. Great Lakes Runoff Intercomparison Project Phase 3: Lake Erie (GRIP-E). Journal of Hydrologic Engineering, pages 1–26, 2021.

- S. L. Markstrom, L. E. Hay, and M. P. Clark. Towards simplification of hydrologic modeling: identification of dominant processes. Hydrology and Earth System Sciences, 20(11):4655–4671, 2016.
- H. McMillan, M. Gueguen, E. Grimon, R. Woods, M. P. Clark, and D. E. Rupp. Spatial variability of hydrological processes and model structure diagnostics in a 50 km² catchment. Hydrological Processes, 28:4896–4913, Aug. 2014.
- H. K. McMillan, M. P. Clark, W. B. Bowden, M. Duncan, and R. A. Woods. Hydrological field data from a modeller’s perspective: Part 1. Diagnostic tests for model structure. Hydrological Processes, 25:511–522, Jan. 2011.
- C. B. Menard, R. Essery, G. Krinner, G. Arduini, P. Bartlett, A. Boone, C. Brutel-Vuilmet, E. Burke, M. Cuntz, Y. Dai, B. Decharme, E. Dutra, X. Fang, C. Fierz, Y. Gusev, S. Hagemann, V. Haverd, H. Kim, M. Lafaysse, T. Marke, O. Nasonova, T. Nitta, M. Niwano, J. Pomeroy, G. Schädler, V. Semenov, T. Smirnova, U. Strasser, S. Swenson, D. Turkov, N. Wever, and H. Yuan. Scientific and human errors in a snow model intercomparison. Bulletin of the American Meteorological Society, pages 1–46, Sept. 2020.
- P. A. Mendoza, M. P. Clark, M. Barlage, B. Rajagopalan, L. Samaniego, G. Abramowitz, and H. Gupta. Are we unnecessarily constraining the agility of complex process-based models? Water Resources Research, 51(1):716–728, Jan. 2015.
- D. N. Moriasi, J. G. Arnold, M. W. Van Liew, R. L. Bingner, R. D. Harmel, and T. L. Veith. Model Evaluation Guidelines for Systematic Quantification of Accuracy in Watershed Simulations. Transactions of the ASABE, 50(3):885–900, 2007.
- J. E. Nash and J. V. Sutcliffe. River flow forecasting through conceptual models: Part I - A discussion of principles. Journal of Hydrology, 10:282–290, 1970.
- R. Nijzink, S. Almeida, I. G. Pechlivanidis, R. Capeli, D. Gustafssons, B. Arheimer, J. Parajka, J. Freer, D. Han, T. Wagener, R. R. P. van Nooijen, H. H. G. Savenije, and M. Hrachowitz. Constraining Conceptual Hydrological Models With Multiple Information Sources. Water Resources Research, 54:8332–8362, Nov. 2018.
- M. Pfannerstill, B. Guse, and N. Fohrer. Smart low flow signature metrics for an improved overall performance evaluation of hydrological models. Journal of Hydrology, 510:447–458, Mar. 2014.
- O. Rakovec, A. H. Weerts, J. Sumihar, and R. Uijlenhoet. Operational aspects of asynchronous filtering for flood forecasting. Hydrology and Earth System Sciences, 19:2911–2924, June 2015.
- M. S. Raleigh, J. D. Lundquist, and M. P. Clark. Exploring the impact of forcing error characteristics on physically based snow simulations within a global sensitivity analysis framework. Hydrology and Earth System Sciences, 19:3153–3179, July 2015.
- M. C. Rogelis and M. Werner. Streamflow forecasts from WRF precipitation for flood early warning in mountain tropical areas. Hydrology and Earth System Sciences, 22(1):853–870, 2018. doi: 10.5194/hess-22-853-2018.
- A. Saltelli, M. Ratto, T. Andres, F. Campolongo, J. Cariboni, D. Gatelli, M. Saisana, and S. Tarantola. Global Sensitivity Analysis. The Primer. Wiley-Interscience, Feb. 2008.
- A. Saltelli, K. Aleksankina, W. Becker, P. Fennell, F. Ferretti, N. Holst, S. Li, and Q. Wu. Why so many published sensitivity analyses are false: A systematic review of sensitivity analysis practices. Environmental Modelling & Software, 114:29–39, 2019.
- B. Schaefli and H. V. Gupta. Do Nash values have value? Hydrological Processes, 21:2075–2080, June 2007.

- A. Snowdon. Improved Numerical Methods for Distributed Hydrological Models. PhD thesis, University of Waterloo, Jan. 2010.
- B. Széles, J. Parajka, P. Hogan, R. Silasari, L. Pavlin, P. Strauss, and G. Blöschl. The Added Value of Different Data Types for Calibrating and Testing a Hydrologic Model in a Small Catchment. Water Resources Research, 56:e2019WR026153, Oct. 2020.
- E. Todini. A model conditional processor to assess predictive uncertainty in flood forecasting. International J. River Basin Management, 6(2):123–137, June 2008.
- A. H. Weerts and G. Y. H. El Serafy. Particle filtering and ensemble Kalman filtering for state updating with hydrological conceptual rainfall-runoff models. Water Resources Research, 42:W09403, Aug. 2006.
- I. Yucel, A. Onen, K. Yilmaz, and D. Gochis. Calibration and evaluation of a flood forecasting system: Utility of numerical weather prediction model, data assimilation and satellite-based rainfall. Journal of Hydrology, 523:49 – 66, 2015. ISSN 0022-1694. doi: <https://doi.org/10.1016/j.jhydrol.2015.01.042>.

Reviewers' Comments:

Reviewer #1:

Remarks to the Author:

Thank you for giving me the opportunity to review this manuscript. The authors did an excellent job in addressing my comments; therefore, I do not have any further comments for the authors.

Reviewer #3:

Remarks to the Author:

All of my concerns have been satisfactorily addressed.